# Dynamics of polarization-tuned mirror symmetry breaking in a rotationally symmetric system

Yu Zhang [1,2,4], Zhibin Li[1,2,4], Zhen Che[3], Wang Zhang[1,2], Yusen Zhang[1,2], Ziqi Lin[1,2], Zhan Lv[1,2], Chunling Wu[1,2], Longwei Han[1,2], Jieyuan Tang[1,2], Wenguo Zhu[1,2], Yi Xiao[1,2], Huadan Zheng[1,2], Yongchun Zhong[1,2], Zhe Chen[1,2] & Jianhui Yu [1,2] ✉

Lateral momentum conservation is typically kept in a non-absorptive rotationally symmetric system through mirror symmetry via Noether's theorem when illuminated by a homogeneous light wave. Therefore, it is still very challenging to break the mirror symmetry and generate a lateral optical force (LOF) in the rotationally symmetric system. Here, we report a general dynamic action in the $SO(2)$ rotationally symmetric system, originating from the polarization-tuned mirror symmetry breaking (MSB) of the light scattering. We demonstrate theoretically and experimentally that MSB can be generally applied to the $SO(2)$ rotationally symmetric system and tuned sinusoidally by polarization orientation, leading to a highly tunable and highly efficient LOF ($9.22 \, \text{pN/mW/µm}^{-2}$) perpendicular to the propagation direction. The proposed MSB mechanism and LOF not only complete the sets of MSB of light-matter interaction and non-conservative force only using a plane wave but also provide extra polarization manipulation freedom.

Symmetry plays an essentially important role in physical systems[1,2]. Noether's theorem indicates that a specific kind of symmetry is closely associated to the conservation of a physical quantity. For instance, in a non-absorptive system, the translational symmetry breaking of the light-matter system through the exchange of the linear optical momentum gives rise to a push or pull optical force[3-9]. The rotational symmetry breaking originates from the exchange of optical angular momentum, leading to an optical torque[10-14]. The gradient force[15-18], a traditional lateral optical force (LOF) perpendicular to the propagation direction, arises from the mirror symmetry breaking (MSB) due to the lateral inhomogeneity of the optical field. Symmetry breaking in the light-matter interaction results in optically dynamic action, supporting optical manipulation. Especially, combining translational symmetry breaking and MSB of the light-matter interaction, Ashkin et al.[19]

pioneered the optical trapping force and has been an essential tool for optical manipulation and biological applications over the past half-century[20-24].

The various MSB mechanisms for the LOF have recently been intensively investigated due to their potential applications in physics and chemistry[25-57]. Technically, it is relatively easy to generate LOF by the objects with intrinsic MSB. The asymmetric geometry of the object, such as a semicylinder rod[25,26], is one of the most intuitive MSB mechanisms that induce a laterally asymmetrical scattering and LOF. Additionally, the chirality of the object could inherently induce MSB through electric-magnetic field coupling, which attracts a great deal of interest due to its potential applications in enantiomer sorting[27-29]. It has been reported that the chirality-dependent LOF can be generated on the chiral object at the air-water interface[27,28] or near the dielectric

[1]Key Laboratory of Optoelectronic Information and Sensing Technologies of Guangdong Higher Education Institutes, Department of Optoelectronic Engineering, Jinan University, Guangzhou 510632, China. [2]Guangdong Provincial Key Laboratory of Optical Fiber Sensing and Communications, Department of Optoelectronic Engineering, Jinan University, Guangzhou 510632, China. [3]Guangdong Science and Technology Infrastructure Center, Guangzhou 510033, China. [4]These authors contributed equally: Yu Zhang, Zhibin Li. ✉e-mail: jianhuiyu@jnu.edu.cn

surface[30]. Recently, an enhanced chirality-dependent LOF has been theoretically predicted through the resonance of multipoles in a specific chiral nanocylinder[31]. The chirality-dependent LOF in the chiral dimer is also theoretically reported, which originates from the optical potential gradient[32,33]. Except for the chiral objects, the chirality-independent LOF is also theoretically predicted to exert on an isotropic spherical particle near the reciprocal[34] or magneto-optical[35] substrates with MSB.

For an $SO(2)$ rotationally symmetric system, it is still challenging to break the mirror symmetry of the light scattering and generate an LOF only using a homogeneous and symmetric optical field. Up to now, several types of LOFs have been achieved by the MSB of light-matter interaction. The spin-orbit interaction excited by a circularly polarized light is a ubiquitous MSB mechanism capable of generating an LOF on an achiral spherical particle near the surface[36–40]. Belinfante's spin momentum[41] (proportional to the curl of spin momentum), as a component of the Poynting momentum, has been confirmed to be an observable MSB mechanism for the LOF on an achiral spherical particle[42–54]. Such MSB mechanism and LOF from the extraordinary Belinfante's spin momentum could arise from the inhomogeneous spin momentum distribution[42], evanescent wave[43–45], tightly focused beam[46,47], two-wave interference[48], unpolarized light[49], surface plasmon polaritons[50,51], and structured light[52,53]. Alternatively, an anomalous MSB mechanism that breaks the mirror symmetry by the electric-magnetic symmetry breaking and generates an LOF is theoretically predicted[55]. It also experimentally demonstrates that a linearly polarized (LP) plane wave could induce multipolar interplay on an achiral elongated nanoparticle to generate LOF[56]. Recently, a polarization-induced LOF has been achieved on the achiral dimer by a simple LP wave, the magnitude and sign of LOF are controllable by flexibly adjusting the orientation of linear polarization[57]. However, the low efficiency ($10^{-2}$ pN/mW/$\mu m^{-2}$)[37,42,45,54] and the special requirements for the MSB, including chiral or structured substrates, chiral materials, complex structured light and scatterer, make the LOF mechanism lose generality and heavily restrict the application of LOF.

Here, we present a general dynamic action in the $SO(2)$ rotationally symmetric system, originating from the polarization-tuned MSB of the light scattering, as depicted in Fig. 1. Theoretical and experimental demonstrates that the MSB can be universally applied to the $SO(2)$ rotationally symmetric system and tuned sinusoidally by polarization orientation, leading to a highly tunable and highly efficient LOF perpendicular to the propagation direction. Importantly, such MSB and LOF mechanisms rely solely on diagonal polarization and incident angle without requiring complex structured light, substrates and special chiral materials. The dual-dipole model is established to reveal that the MSB and LOF stem from the scattering interaction between the two dipoles. Furthermore, we experimentally observed the dynamic action of a dodecane oil droplet semi-floating at the air-water interface and confirmed the sinusoidal dependence of the LOF on the polarization angle. Intriguingly, the LOF efficiency of 9.22 pN/mW/$\mu m^{-2}$ is achieved in the experiment, which is at least two orders higher than that previously reported and opens the way to exploit the polarization for optical manipulation of mirror-symmetric objects.

## Results
### Principle of polarization-induced mirror symmetry breaking in an $SO(2)$ rotationally symmetric system
Figure 1 shows the scheme for the LOF arising from polarization-induced MSB in an $SO(2)$ rotationally symmetric system. In Fig. 1a, an LP plane wave ($\lambda = 532$ nm) is obliquely incident on a rotationally symmetric object at an incident angle $\theta$ with respect to the rotational axis ($z$-axis). The polarization angle $\alpha$ of the LP plane wave is defined as the angle between the optical field $\mathbf{E}$ and the $+y$ direction. The five common objects with $SO(2)$ rotationally symmetry, including a hemisphere, a cone, a cylinder, a dimer, and a lens, are shown in the inset of

Fig. 1a. Considering the hemisphere as a typical example, at a diagonal polarization (such as $\alpha = 45°$), the mirror symmetry of the light scattering with respect to the $xz$-plane will be broken and the lateral scattering momentum $P_{-y}$ in the $-y$ direction is greater than the lateral scattering momentum $P_{+y}$ in the $+y$ direction. Consequently, the lateral momentum conservation gives rise to an LOF along the $+y$ direction exerting on the mirror-symmetric object, as shown in Fig. 1b. For the $s$- ($\alpha = 0°$) or $p$- ($\alpha = 90°$) polarization, as shown in Fig. 1c, d, the light-matter interaction holds mirror symmetry with respect to the $xz$-plane and the lateral scattering optical momentum remains conserved in the $y$ direction without the transfer of the lateral momentum to the object, thus leading to a vanishing LOF. Recently, the LOF exerted on a cylindrical Ag nanowire demonstrated by Nan et al. using an obliquely incident diagonal polarization light can be regarded as the case of the special cylinder in our proposed MSB mechanism[56]. It should be noticed that, as shown in the following, the dual-dipole interaction instead of higher multipolar interaction can fully explain the origin of the LOF exerted on the cylindrical system. Additionally, we also notice that the mirror asymmetry of the light-matter system with respect to the $xy$-plane is an essential prerequisite for breaking the mirror symmetry about the $xz$-plane and generating the LOF by diagonal polarization. For a mirror-symmetric object about the $xy$-plane (e.g., cylinder), the oblique incident of light ($0° < \theta < 90°$) capable of breaking the $xy$-plane mirror symmetry becomes an essential condition to break the $xz$-plane mirror symmetry and generate the LOF in the $y$ direction. However, under the vertical incidence of light ($\theta = 90°$), the diagonal polarization could not break the $xz$-plane mirror symmetry and the LOF vanishes. Contrarily, when an object is mirror-asymmetric about the $xy$-plane (e.g., hemisphere or cone), even under the vertical incidence of light, the diagonal polarization could still break the mirror symmetry with respect to the $xz$-plane and generate the LOF along the $y$ direction (Supplementary Note 1).

To corroborate such polarization-induced MSB and LOF in the $SO(2)$ rotationally symmetric system, the scattering near-fields in the $yz$-plane of five different rotationally symmetric objects were numerically calculated using the 3D finite-difference time-domain (3D-FDTD) (see Methods). The five kinds of common rotationally symmetric objects are made of isotropic polystyrene (PS) and placed in air. Since the lateral component of the Poynting vector is closely related to the lateral optical momentum and the LOF, the scattering near-field $|E_z|$ in the $yz$-plane ($x = 0$) was plotted in Fig. 1e–i under the polarization angle $\alpha = 45°$ and incident angle $\theta = 60°$. The green arrow in the figure denotes the linear polarization orientation. It is clearly seen that the scattering field exhibits mirror asymmetry with respect to the $xz$-plane at a diagonal polarization. The asymmetrical light scattering breaks the conservation of the lateral optical momentum in the $y$ direction, transferring the lateral optical momentum to the particle and thus generating an LOF $F_y$ acting on the particle. Using the scattering momentum, the LOF can be phenomenally expressed as $F_y = -\frac{D_{MSB}}{c}\gamma C_{sca}^{total} I_{inc}$. Here, $I_{inc}$ is the intensity of the incident plane wave. The total scattering cross-section of Minkowski momentum $C_{sca}^{total} = n_1 C_{sca}^{air}$, where $C_{sca}^{air}$ is the scattering cross-section of the particle in the air. The degree of MSB, $D_{MSB}$, is defined as the ratio of the difference of the scattering momentum along the $+y$ direction and the $-y$ direction to the total scattering momentum along the $y$ direction, and $\gamma$ is the ratio of the total scattering momentum in the $y$ direction to the total scattering momentum in all directions (see Methods). From the numerical simulation, we obtained the dependence of $D_{MSB}$ on polarization angle $\alpha$ and shown in Fig. 1j. It is seen that the $D_{MSB}$ varies sinusoidally with the polarization angle $\alpha$, and reaches the maximum $D_{MSB}$ at $\alpha = 45°$. The corresponding LOF was calculated by integrating the Maxwell stress tensor, as shown in Fig. 1k. For arbitrary objects with $SO(2)$ continuous rotational symmetry, the LOF $F_y$ exhibits a sinusoidal dependence of $\sin(2\alpha)$. The sinusoidal dependence is attributed to the

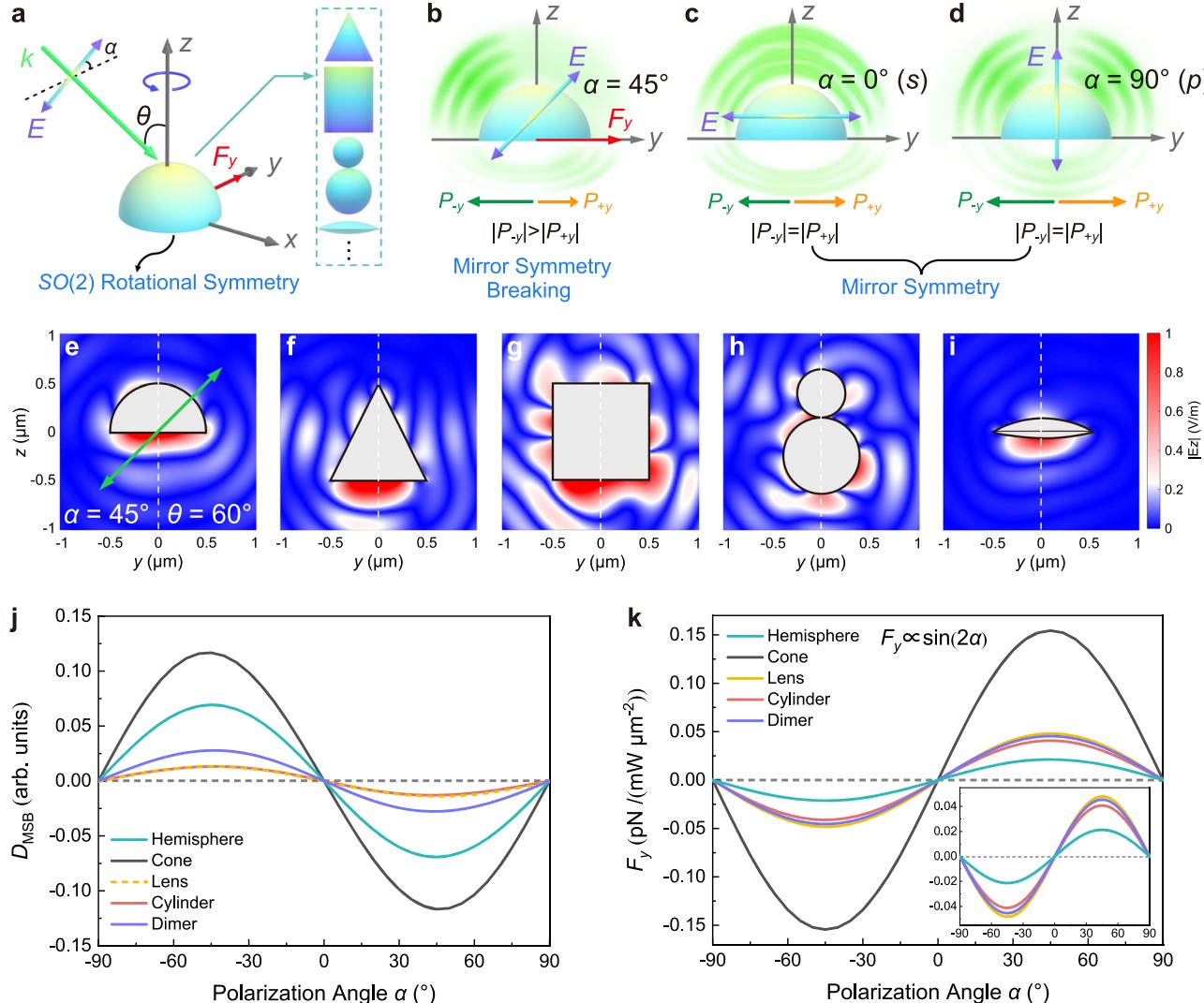

**Fig. 1 | Scheme for the lateral optical force arising from polarization-induced mirror symmetry breaking in an *SO*(2) rotationally symmetric system.** **a** Schematic illustration of the lateral optical force acting on an object with *SO*(2) continuous rotational symmetry. A linearly polarized plane wave with a polarization angle $\alpha$ obliquely illuminates onto the object at an incident angle $\theta$. The objects have only one axis of rotational symmetry, such as a cone, a cylinder, a dimer and a lens, etc., as shown in the inset. **b** At a diagonal polarization (such as $\alpha = 45°$), the mirror symmetry with respect to the *xz*-plane of the light-matter interaction is broken and the lateral scattering optical momentum $P_{-y}$ in the -y direction is greater than the lateral scattering optical momentum $P_{+y}$ in the +y direction. Consequently,

the lateral total momentum conservation results in a lateral optical force $F_y$ along the +y direction. **c**, **d** For a *s*-or *p*-polarized plane wave ($\alpha = 0°$ or $\alpha = 90°$), the light-matter interaction holds mirror symmetry, thus yielding a vanishing lateral optical force $F_y$. **e**–**i** The scattering near-field $|E_z|$ in the *yz*-plane (x = 0) of the five *SO*(2) rotationally symmetric objects at the polarization angle $\alpha = 45°$, incident angle $\theta = 60°$. **j** The degree of mirror symmetry breaking ($D_{\mathrm{MSB}}$) of the five objects depends sinusoidally on the polarization angle $\alpha$. **k** Numerically calculated lateral optical force $F_y$ shows a sin(2$\alpha$) dependence on the polarization angle $\alpha$ for the *SO*(2) rotationally symmetric objects.

dipole interaction terms as predicted in the following Eqs. (4) and (5). The proposed MSB mechanism allows the linear polarization to sinusoidally tune the degree of MSB of light scattering, providing an efficient way to tune the amplitudes and directions of the LOF.

To understand intuitively the extraordinary MSB mechanism and LOF, we consider a specific system with *SO*(2) continuous rotational symmetry that could be easily implemented by experiment. The system is schematically shown in Fig. 2a. Here, an isotropic dielectric spherical particle with the highest symmetry is semi-floating at the interface of two mediums, half part of the particle in medium 1 (air, refractive index $n_1$) and the other half in medium 2 (water, refractive index $n_2$). An LP plane wave ($\lambda = 532$ nm) with a polarization angle $\alpha$ illuminates obliquely on the particle from the air at an incident angle $\theta$. Such a strategy of utilizing the buoyancy at the interface is widely used in applications such as chiral enantiomeric sorting, medical diagnostics and analytical chemistry[27,28,31,37,39,58,59].

Numerical simulation is performed using the 3D-FDTD to extract the near-field scattering field near the particle. In the calculation, a PS particle ($R = 500$ nm, $n_p = 1.5983$ @ $\lambda = 532$ nm) is semi-floating at the air-water interface, which is obliquely illuminated by a single LP plane wave at an incident angle of $\theta = 75°$. The scattering near-field $|E_z|$ in the *yz*-plane is shown in Fig. 2b–e for the polarization angles $\alpha = -45°$, 90°, 45° and 0°, respectively. It is seen that the pattern and the mirror symmetry of the scattering near-field vary with the polarization orientation of the incident light. In Fig. 2b (Fig. 2d), when illuminated by the LP plane wave with a polarization angle $\alpha = -45°$ (45°), the scattering near-field $|E_z|$ exhibits an obvious MSB. The scattering field irradiates mainly along a direction nearly perpendicular to the polarization of the incident light and exhibits a stronger scattering in the air than in the water due to the much larger refractive index contrast ($\Delta n = n_p - n_1$) of the particle in the air, thus leading to a strong MSB of the light scattering. The scattering field in Fig. 2c, e ascertain that the

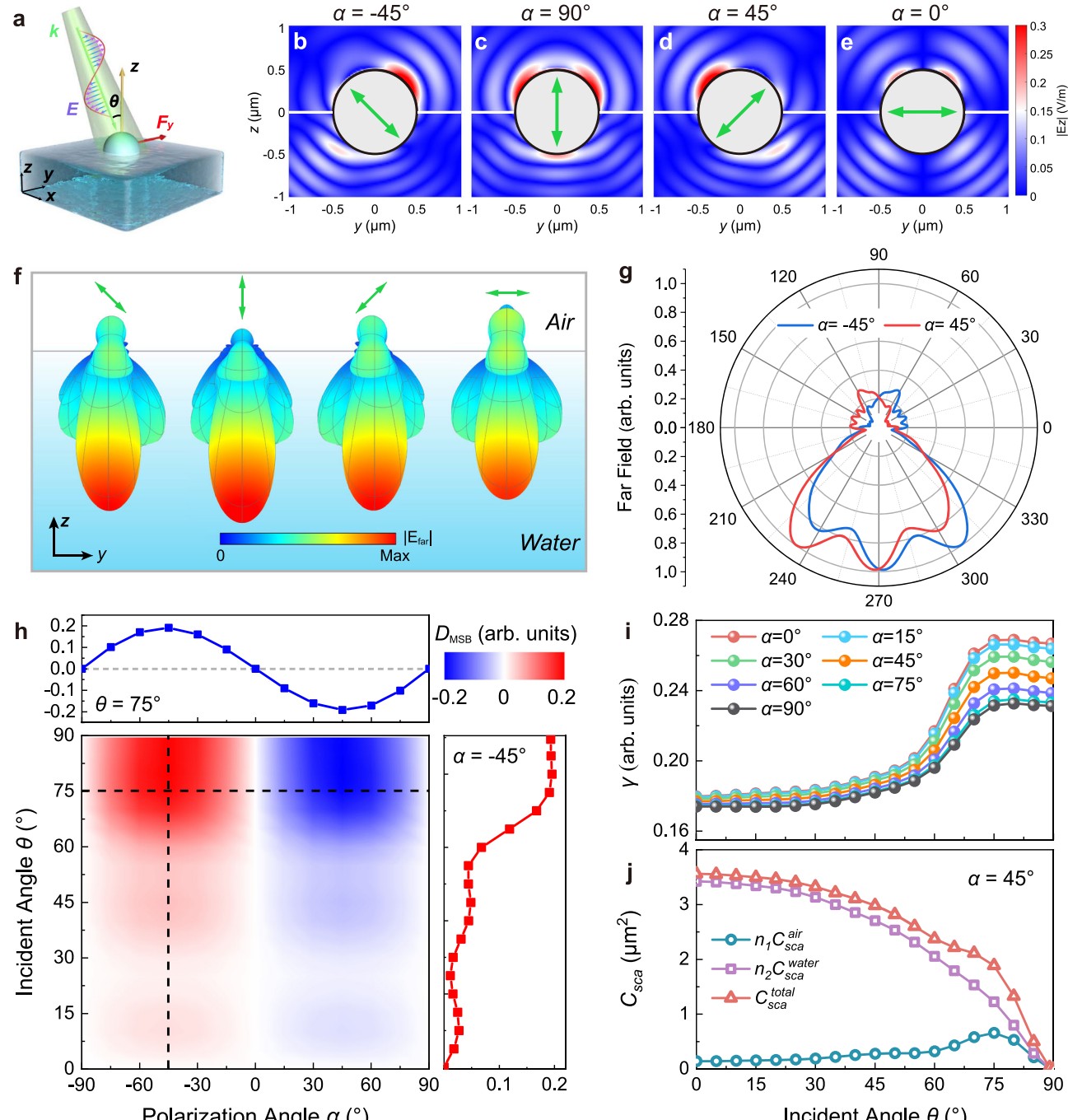

**Fig. 2 | Numerical simulation of the polarization-induced mirror symmetry breaking. a** Schematic illustration of the lateral optical force acting on an isotropic spherical particle semi-floating at the air-water interface. A 532 nm linearly polarized plane wave with a polarization angle $\alpha$ obliquely illuminates onto the particle at an incident angle $\theta$. **b**–**e** The scattering near-field $|E_z|$ in the $yz$-plane (x = 0) at polarization angles $\alpha = -45°$, 90°, 45° and 0°, respectively. The white circles denote the spherical particle and the green arrows in the center denote the polarization orientations. The polystyrene particle radius is 500 nm and the incident angle is $\theta = 75°$. **f** Solid-angular dependence of the far-field radiation. **g** Azimuthal dependence of the far-field in the $yz$-plane extracted from (**f**) for polarization angle $\alpha = \pm 45°$. **h** The degree of mirror symmetry breaking ($D_{MSB}$) dependence on the polarization angle $\alpha$ and incident angle $\theta$. The top blue line shows a $\sin(2\alpha)$ dependence of the $D_{MSB}$ on polarization angle $\alpha$ at an incident angle of $\theta = 75°$. The red line on the right shows the variation of $D_{MSB}$ with the incident angle $\theta$ when the polarization angle $\alpha = -45°$. **i** The ratio $\gamma$ of the total scattered momentum in the $y$ direction to the total scattered momentum in all directions as a function of the angle of incidence. **j** The dependence of the scattering cross-section $C_{sca}$ of the semi-floating particle on the incident angle $\theta$ at the polarization angle $\alpha = 45°$.

light scattering turns back to be mirror-symmetrical and the LOF vanishes when the polarization is tuned to $p$- ($\alpha = 90°$) or $s$- ($\alpha = 0°$) polarization. Notice that when the particle is immersed fully in the water or fully in the air, the diagonal polarization of the incident field hardly breaks mirror symmetry and generates the LOF (Supplementary Fig. 2). It reveals that the semi-floating of the particle at the air-water interface contribute to enhance the degree of MSB and the LOF efficiency, attributed to the configuration is mirror symmetry broken about the $xy$-plane. Consequently, the polarization-induced MSB of light scattering from the semi-floating particle not only provides a highly efficient mechanism for the LOF but also a highly tunable method for the amplitude and direction of the LOF.

To investigate the MSB of the light scattering more rigorously, the solid-angle dependence of the far field was calculated using the 3D full-vector finite element method (3D-FVFEM) and visualized in the spherical coordinates in Fig. 2f. The azimuthal dependence of the far field in the $yz$-plane is also extracted from Fig. 2f and plotted in polar coordinator in Fig. 2g. It is solidly verified that the linear polarizations $\alpha = \pm 45°$ give rise to the MSB of the particle scattering. According to the phenomenological expression for the LOF, all of the parameters ($D_{MSB}$, $\gamma$ and $C_{sca}^{total}$) together determine the normalized LOF (LOF efficiency) $F_y/I_{inc}$ and become critical to improving the LOF efficiency. These parameters, as shown in Fig. 2h–j, was calculated through numerical simulation. Figure 2h shows the degree of MSB ($D_{MSB}$) dependence on the polarization angle $\alpha$ and the incident angle $\theta$. It is shown by the red line on the right of Fig. 2h that, for a polarization angle $\alpha = -45°$, the $D_{MSB}$ increases from 0 to 0.2 when the incident angle $\theta$ increases from 0° to 75° with a little fluctuation, while the $D_{MSB}$ reaches up to a maximum of 0.2 in the range of 75° ~ 90°. This indicates that the larger incident angle $\theta$ in the range of 75° ~ 90° could induce the stronger MSB of the scattering and the larger LOF. Intuitively, $D_{MSB}$ will be vanishing at incident angle $\theta = 0°$ regardless of the polarization angle due to the azimuthal symmetry. Additionally, the amplitude and sign of the $D_{MSB}$ can be tunable by the polarization angle $\alpha$ ranging from −90° to 90°, exhibiting a $\sin(2\alpha)$ dependence as shown by the blue line at the top of Fig. 2h. For a given incident angle, the $D_{MSB}$ can reach up to a maximum at $\alpha = \pm 45°$, indicating that the strongest MSB of light scattering and the strongest LOF are both achieved at $\alpha = \pm 45°$.

The scattering ratio $\gamma$ also plays an important role in improving LOF efficiency. In Fig. 2i, the $\gamma$ increases slowly at the incident angle of 0° ~ 55° and increases significantly at 60° ~ 90° at a given polarization angle. This indicates that more scattering optical momentum is concentrated in the $y$ direction to induce a stronger LOF at the large incident angles than at the small incident angles. In addition, for a given incident angle, the ratio $\gamma$ gradually decreases as the polarization angle $\alpha$ changes from 0° to 90°. That is, the ratio $\gamma$ of the scattering momentum in the $y$ direction achieves the largest at polarization angle $\alpha = 0°$. However, Fig. 2h shows that $D_{MSB} = 0$ when $\alpha = 0°$. Consequently, the LOF $F_y$ becomes vanishing due to the vanishing MSB even when $\gamma$ achieves maximum at $\alpha = 0°$. Besides, the total scattering cross-section $C_{sca}^{total}$ is another important factor for the LOF efficiency. The $C_{sca}^{total}$ and $C_{sca}^{air(water)}$ are calculated by the FVFEM and their dependence on $\theta$ is shown in Fig. 2j. It is seen in Fig. 2j that both $C_{sca}^{water}$ and $C_{sca}^{total}$ gradually decrease as the incident angle $\theta$ increases from 0° to 90°, and finally tends to 0 μm$^2$ at $\theta = 90°$, while $C_{sca}^{air}$ increases up to a maximum of 0.66 μm$^2$ at $\theta = 75°$ and decreases down to 0 μm$^2$ at $\theta = 90°$. Although the polarization orientation could induce the strongest MSB of the light scattering at $\alpha = \pm 45°$, $\theta = 90°$, the vanishing $C_{sca}^{total}$ at $\theta = 90°$ makes the LOF zero. Considering the proportion of the LOF to the product of the parameters $D_{MSB}$, $\gamma$ and $C_{sca}^{total}$, it is concluded that the LOF should be up to maximum at the incident angle of $\theta = 75°$ as both of $D_{MSB}$ and $\gamma$ achieve maximum. The maximum LOF prediction is consistent with the following FDTD calculated LOF. It is interesting to notice that the total scattering cross-section $C_{sca}^{total}$ is down to 1.9 μm$^2$ at $\theta = 75°$, which is 7.6 times larger than the cross-section (0.25 μm$^2$) of the 0.5 μm radius sphere. The larger $C_{sca}^{total}$ is attributed to the enhancement of scattering by the reflection at the air-water interface at a large incident angle. Correspondingly, the LOF could be enhanced largely by the larger scattering cross-section.

## Dual-dipole model for polarization-induced mirror symmetry breaking

According to the concept of the discrete dipole approximation[60], the object with $SO(2)$ rotational symmetry could be approximately simplified as two spherical dipoles arrayed along a rotational axis. Therefore, to unravel the origin of the polarization-induced MSB and the LOF, a dual-dipole theoretical model was established to derive

the LOF analytically in the dipole approximation ($R\ll\lambda$). The semi-floating nano-particle at the air-water interface could be approximatively considered as two electric dipoles[31,37]. Here, dipole 1 at $\mathbf{r}_1 = (x_0, y_0, z_1)$ and dipole 2 at $\mathbf{r}_2 = (x_0, y_0, z_2)$ are respectively located in semi-infinite transparent medium 1 and medium 2 (Supplementary Fig. 3a). The time-averaged LOF acting on an electric dipole can be generally expressed as $\langle F_y \rangle = (1/2)\mathrm{Re}\left(\mathbf{d}\partial_y\mathbf{E}^*\right)$[37,61], where $\mathbf{d}$ is the induced electric dipole moment, and $\mathbf{E}$ is the local field at the location of the dipole. For dipole 1, the local field is $\mathbf{E}_1 = \mathbf{E}_{inc} + \mathbf{E}_{ref} + \mathbf{E}_{d21}$, and the local field is $\mathbf{E}_2 = \mathbf{E}_{tr} + \mathbf{E}_{d12}$ for dipole 2. The fields $\mathbf{E}_{inc}$, $\mathbf{E}_{ref}$, $\mathbf{E}_{tr}$ and $\mathbf{E}_{dij}$ denote the incident, reflected, transmitted and scattering fields, respectively. The subscripts in the scattering field denote the field located at the dipole $j$ scattered from dipole $i$ (Supplementary Fig. 3b). Since the higher-order light scattering in the dual-dipole model is neglected for its ultra-weakness, the dual-dipole interaction through the first-order scattering fields $\mathbf{E}_{d12}$ and $\mathbf{E}_{d21}$ were only considered in the model for the LOF. The total LOF induced by the scattering interaction of the two dipoles was derived from the sum of the LOFs acting on the two dipoles (Supplementary Note 3)

$$\langle F_y \rangle = \frac{k_0^2}{\varepsilon_0}[\mathrm{Re}(\partial_y\overleftrightarrow{G}_{1,yz}^{tr})\mathrm{Re}(d_{2y}d_{1z}^*) + \mathrm{Re}(\partial_y\overleftrightarrow{G}_{2,yz}^{tr})\mathrm{Re}(d_{1y}^*d_{2z})] \quad (1)$$

$$\partial_y\overleftrightarrow{G}_{1,yz}^{tr} = -\frac{1}{8\pi k_1 k_2}\int_0^\infty \frac{k_{z2}}{k_{z1}}t_1^p(k_\rho)k_\rho^3 \exp[i(k_{z1}z_1 - k_{z2}z_2)]dk_\rho \quad (2)$$

$$\partial_y\overleftrightarrow{G}_{2,yz}^{tr} = -\frac{1}{8\pi k_1 k_2}\int_0^\infty \frac{k_{z1}}{k_{z2}}t_2^p(k_\rho)k_\rho^3 \exp[i(k_{z1}z_1 - k_{z2}z_2)]dk_\rho \quad (3)$$

Here, $\overleftrightarrow{G}_{1,yz}^{tr} = \overleftrightarrow{G}_{yz}^{tr}(\mathbf{r}_2, \mathbf{r}_1)$ and $\overleftrightarrow{G}_{2,yz}^{tr} = \overleftrightarrow{G}_{yz}^{tr}(\mathbf{r}_1, \mathbf{r}_2)$ are the $yz$ components of the transmitted Green's function tensors for dipole 1 and dipole 2, respectively. In Eqs. (1)–(3), $\varepsilon_0$ is the vacuum dielectric permittivity, $k_1$ and $k_2$ are wavevectors in media 1 and 2, $k_{zi} = \sqrt{(k_i^2 - k_\rho^2)}$ ($i = 1,2$), and $k_0$ is the wavenumber in a vacuum. Here, $t_1^p$ and $t_2^p$ are the Fresnel transmission coefficients for the $p$-polarized wave in media 1 and 2, respectively. The $d_{mn}$ ($m = 1, 2$, $n = z, y$) is the dipole moment component $n$ of the $m$-th dipole. When the LP plane wave illuminates from the less dense medium (i.e., air, $n_1 = 1$) to the denser medium (i.e., water, $n_2 = 1.337$), the dipole moment term in Eq. (1) can be expressed as (Supplementary Note 4)

$$\mathrm{Re}(d_{2y}d_{1z}^*) = \frac{1}{2}\alpha_{e,1}\alpha_{e,2}|E_{inc}|^2\sin(2\alpha)\psi_1(\theta, z_1) \quad (4)$$

$$\mathrm{Re}(d_{1y}^*d_{2z}) = \frac{1}{2}\alpha_{e,1}\alpha_{e,2}|E_{inc}|^2\sqrt{\varepsilon_{12}}\sin(2\alpha)\psi_2(\theta, z_1) \quad (5)$$

Here $\alpha_{e,i}$ is electric polarizability for the dipole in medium $i$, and the dipole is assumed as a non-absorptive dielectric particle, $\mathrm{Im}(\alpha_e) = 0$. $\varepsilon_{12} = \varepsilon_1/\varepsilon_2$ is the relative dielectric permittivity of the two media. The factors $\psi_1(\theta, z_1) = t_s(\theta)\sin(\theta)[\cos(\delta_{d12}) + r_p(\theta)\cos(\delta_{ref} - \delta_{d12})]$, $\psi_2(\theta, z_1) = t_p(\theta)\sin(\theta)[\cos(\delta_{d12}) + r_s(\theta)\cos(\delta_{ref} - \delta_{d12})]$. The phase $\delta_{ref} = 2k_1 z_1\cos(\theta)$ is the retardation between the incident field $\mathbf{E}_{inc}$ and the reflected field $\mathbf{E}_{ref}$ at dipole 1. The phase $\delta_{d12}$ corresponds to the retardation between the incident field $\mathbf{E}_{inc}$ at dipole 1 and the transmitted field $\mathbf{E}_{tr}$ at dipole 2 (Supplementary Fig. 4). The Eqs. (4) and (5) from the dual-dipole model predict that the dipole moment terms and the LOF should be proportional to $\sin(2\alpha)$, exhibiting the same $\alpha$ dependence as the $D_{MSB}$ in Fig. 2h. The dependence confirms again that the extraordinary LOF originates from the polarization-induced MSB and the dual-dipole scattering interaction. When the incident angle $\theta = 0°$, the LOF vanishes because of the azimuthal symmetry for an arbitrary polarization orientation. The LOF also vanishes at $\theta = 90°$ because no light is transmitted into medium 2 ($t_s = t_p = 0$ at $\theta = 90°$) and

no dual-dipole scattering interaction occurs, agreeing well with the simulation in Fig. 3.

In Eqs. (4) and (5), the induced dipole terms are proportional to the electric polarizability $\alpha_{e,i}$ and the factor $\psi_i(\theta, z_1)$. The factor $\psi_i(\theta, z_1)$ is closely related to the cosine function of the phase retardation $\delta_{d12}$, where $\delta_{d12}$ is proportional to the product between the dipole separation $|z_i|$ ($|z_1| = |z_2|$) and the cosine function of incident angle $\theta$ (Supplementary Eq. (23)). Consequently, the magnitude of $\text{Re}(d_{2y}d_{1z}^*)$ and $\text{Re}(d_{1y}^*d_{2z})$ oscillates with the incident angle and dipole radius $R_d$ (approximately equivalent to the dipole separation $|z_1|$). In particular, their oscillatory variation becomes increasingly rapid with respect to the incident angle with the increase in radius (Supplementary Fig. 4d–f). Furthermore, the oscillatory amplitude of the dipole moment term increases with the dipole radius $R_d$ approximately in $R_d^6$ way in the dipole range (Supplementary Fig. 4g–i). This is contributed to the dipole electric polarizability $\alpha_{e,i}$, where $\alpha_{e,i}$ is described as a $R_d^3$ dependent Clausius-Mossotti relationship, $\alpha_{e,i} = 4\pi\varepsilon_i R_d^3[(\varepsilon_p - \varepsilon_i)/(\varepsilon_p + 2\varepsilon_i)]$. Here, $\varepsilon_p$ and $\varepsilon_i$ are the permittivities of the particle and medium $i$, respectively. Such oscillatory dependence on the radius and incident angles also has an impact on the LOF, as shown in Fig. 3c. Besides, the phase retardation $k_{zi}z_i$ in the integral of the Green's functions also renders the terms of $\partial_y \overset{\leftrightarrow}{G}{}^{tr}_{i,yz}$ ($i = 1, 2$) oscillatory with the radius $R_d$ ($R_d = |z_1|$). The integral of Eqs. (2) and (3) can be evaluated approximately (Supplementary Note 4). When the two dipoles close to the interface between two media ($k_2z_1 \ll 1$) and the high dielectric contrast ($\varepsilon_{12} \ll 1$) is considered, the LOF can be simplified as

$$\langle F_y \rangle \approx -\frac{3k_0^2 k_2^2}{4\pi\varepsilon_0(k_2 z_1)^4}\alpha_{e,1}\alpha_{e,2}|E_{inc}|^2 \sin(2\alpha)\psi_1(\theta, z_1) \qquad (6)$$

It can be clearly seen in Eq. (6) that the LOF varies with the incident angle $\theta$ and the phase retardation $k_2z_1$ in the factor $\psi_1(\theta, z_1)$. As shown in Supplementary Fig. 5, the dependences of LOF on the dipole radius $R_d$ and the incident angle $\theta$ ($\varepsilon_{12} = 1/100$, $\alpha = 45°$) verifies that the approximate expression of Eq. (6) could describe the LOF accurately when $k_2z_1 \to 0$ and $\varepsilon_{12} \ll 1$. Additionally, multiplication of $\text{Re}(\partial_y \overset{\leftrightarrow}{G}{}^{tr}_{i,yz})$ ($\propto 1/R_d^4$, Supplementary Eq. (36)) with the term $\alpha_{e,1}\alpha_{e,2}$ ($\propto R_d^6$) leads to the LOF $\langle F_y \rangle \propto R_d^2$ when $R_d \to 0$ ($k_2z_1 \to 0$), as demonstrated by the quadratic function fitting of the green line in Supplementary Fig. 5a.

## Numerical calculation of normalized lateral optical force

The 3D-FDTD simulation was performed to quantitatively assess the LOF on the Rayleigh and Mie particles semi-floating at the air-water interface. We calculated the normalized LOF (LOF efficiency) versus the polarization $\alpha$ and the incident angle $\theta$ for a 50 nm and a 500 nm radius PS particle, as respectively shown in Fig. 3a, b. One can see that the LOF $F_y$ is proportional to $\sin(2\alpha)$, agreeing well with the prediction of the dual-dipole model by Eqs. (4) and (5). This validates the validity of the dual-dipole model and the LOF origin of the dual-dipole scattering interaction. The detailed dependence of $F_y$ on $\alpha$ for different incident angle $\theta$ is shown at the top of Fig. 3a, b. We notice that the sign of $F_y$ on a 50 nm radius Rayleigh particle is opposite to that on a 500 nm radius Mie particle. This is approximately explained by the phase retardation, which is closely related to the particle size in the integral terms of Eqs. (2) and (3) and the dipole moment terms of Eqs. (4) and (5). The cosine dependence of the phase retardation ($\sim k_1R_d$ or $k_2R_d$) should lead to the cosine oscillatory of the LOF and thus the reversal of LOF at a given radius. Particularly, in the dipole region, the numerical calculation of Eq. (1) in Supplementary Fig. 6g indicates that when the dipole radius in the dual-dipole model is

<72 nm (equivalent to the particle radius <144 nm), a negative LOF will be generated by an LP light with $\alpha > 0°$ (Supplementary Note 5). It merits attention that the dual-dipole model predicts the LOF vanishing at the 144 nm radius of the particle, which is well consistent with the FDTD-calculated LOF at a 150 nm radius in the inset of Fig. 3c. This further manifest that the dual-dipole model could accurately describe the LOF of the dipole particle.

The LOF acting on the particles with different radii also exhibited distinctly different dependence on incident angle $\theta$, as shown in the right of Fig. 3a and Fig. 3b. For a 50 nm radius Rayleigh particle, the LOF rises when incident angle $\theta$ is increased from 0° to 50° and gradually vanishes when incident angle $\theta$ is further increased to 90°. While for a 500 nm radius Mie particle, the LOF oscillates with $\theta$ and reaches a maximum around the incident angle of $\theta = 75°$ at which the $D_{MSB}$ and the ratio $\gamma$ achieve the maximum (see Fig. 2h and Fig. 2i). As the particle radius increases, this oscillatory behavior becomes more pronounced, as shown in Fig. 3c. Figure 3c shows the LOF dependence on the incident angle $\theta$ for the different particle radii ranging from 50 nm to 1000 nm at the polarization angle $\alpha = 45°$. Here, the inset in Fig. 3c shows the LOF acting on the particles with radii ranging from 50 nm to 200 nm. The simulation results authenticate the dual-dipole model prediction that the LOF oscillates more dramatically with the incident angle $\theta$ at a larger radius (Supplementary Fig. 4d–f). The LOF dependence on the particle radius $R$ is further shown in Fig. 3d, where the radius ranges from 300 nm to 1000 nm. It is seen that small incident angles, e.g., 15° ~ 30°, can easily induce the oscillations of the LOF, as shown in the inset, which originates from the separation-dependent phase retardation $\delta_{d12}$ in the dipole moment terms. The geometric analysis could easily show that the phase retardation becomes more sensitive to the dipole separation (particle size) at a smaller incident angle than at a larger incident angle. Therefore, the oscillatory variation of the dipole moment terms with the particle radius becomes more rapid at smaller incident angles while slow at larger ones (Supplementary Fig. 4g–i). As expected, the LOF steadily increases with the particle radius at the larger incident angles of 60° ~ 75°. Additionally, with an increase in the particle radius, the LOF becomes weak and its variation also becomes small at the incident angle range of 0° ~ 45°. This phenomenon could be explained by two factors: (1) the $D_{MSB}$ becomes weak and varies slightly at small incident angles, as shown in Fig. 2h; (2) the ratio $\gamma$ of the scattering momentum in the $y$ direction to the total scattered momentum decreases with the particle radius, despite of the scattering cross-section $C_{sca}^{air(water)}$ proportional to $R^2$. It is also seen that the LOF always disappears for normal incidence ($\theta = 0°$) or grazing incidence ($\theta = 90°$), which is consistent with dual-dipole theoretical predictions.

For a particle in the geometrical optics scale ($R \gg \lambda$), we also analyze the LOF dependence on incident angle $\theta$, polarization angle $\alpha$, and radius $R$, respectively, by using the 3D-Raytracing method (Supplementary Note 6). Especially, the ray trajectory on the $yz$ projection and their intensity distributions at different polarization angles $\alpha$ are shown in Supplementary Fig. 7 and Fig. 9. It is obvious that the mirror symmetry of the scattering field is held at polarization angle $\alpha = 0°$ or 90° and broken at $\alpha = \pm45°$. These findings suggested that, even when the micro-/macro particle is illuminated by an incoherent light, the polarization-induced MSB mechanism still operates to generate LOF, which also exhibits the sinusoidal polarization dependence predicted by the dipole moment terms in Eqs. (4) and (5).

## Experimental observation of lateral optical force

Dodecane oil droplets ($n_p = 1.4216$ @ $\lambda = 532$ nm) with $SO(2)$ rotational symmetry were used to experimentally verify the MSB mechanism and LOF. Due to the interfacial tension, the oil droplet becomes lens-shape[8] and semi-floats at the air-water interface when released on the liquid surface (Supplementary Fig. 9a). The LOF of the lenticular oil droplet was numerically calculated and the LOF also shows a $\sin(2\alpha)$

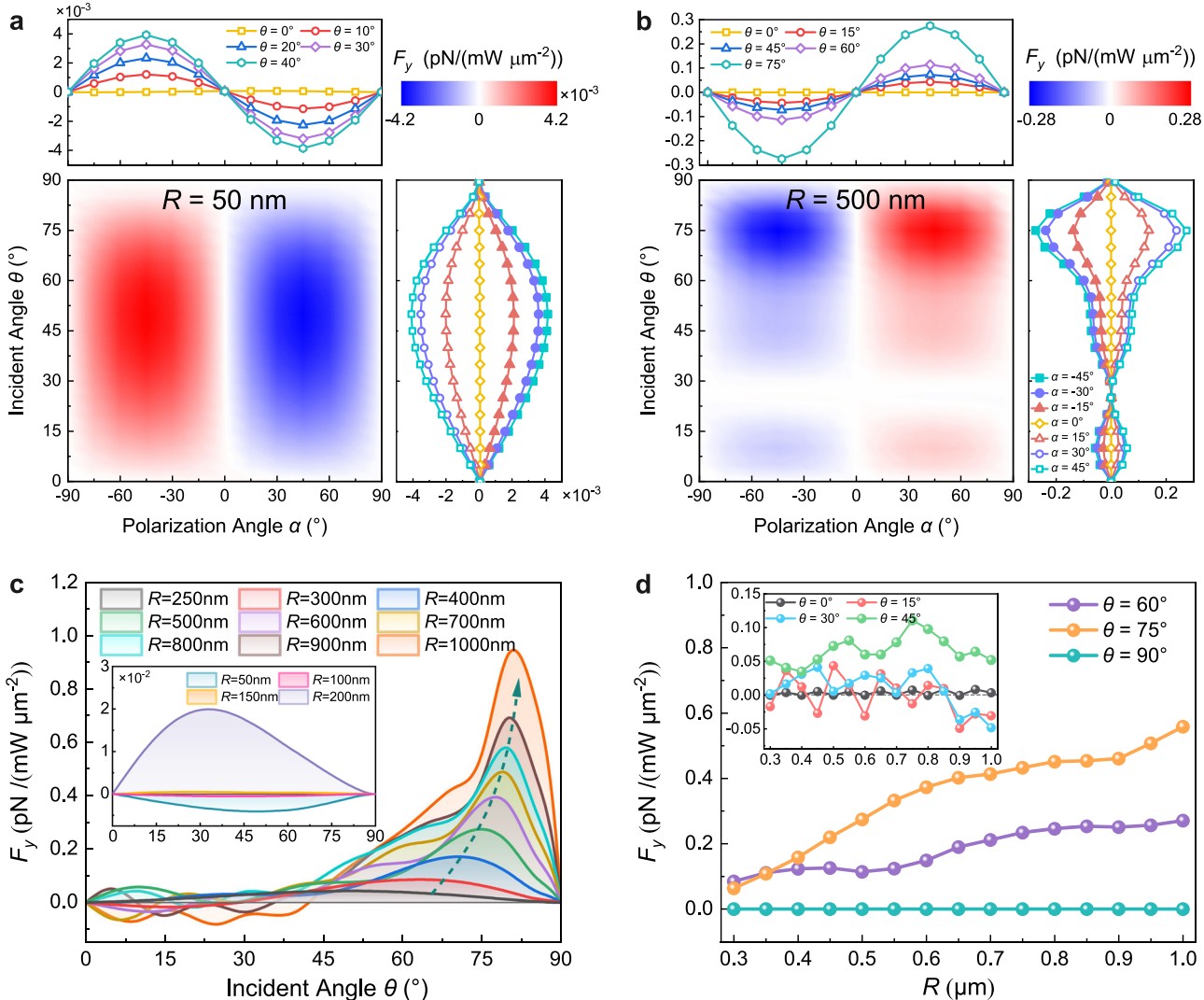

**Fig. 3 | Numerical calculation of normalized lateral optical force on a polystyrene particle semi-floating at the air-water interface. a, b** The normalized lateral optical force dependence on polarization angle $\alpha$ and incident angle $\theta$ for a 50 nm radius (**a**) and a 500 nm radius (**b**) polystyrene particle. **c** The lateral optical force variation with incident angle $\theta$ for different particle radii $R$. **d** Dependence of the lateral optical force on the radius $R$ in Mie range at polarization angle $\alpha = 45°$.

dependence on the polarization angle $\alpha$ and reaches a maximum around the incident angle of $\theta = 55°$ (Supplementary Note 6 and Note 7). The sinusoidal dependence indicates that the oil droplet semi-floating at the interface is another special case of the $SO(2)$ rotationally symmetric system, allowing us to validate the MSB and the LOF safely. The preparation of oil droplets and the experimental scheme are detailed in Methods and Supplementary Note 8. An LP and no-structured laser beam at 532 nm with the power of 1.83 W is focused into a line shape beam by a cylindrical lens, where the beam profile of 100 µm × 900 µm was measured on a glass slide, as shown in Fig. 4a. According to the measured beam profile, the Gaussian waists $w_y$ in the long axes ($y$) and $w_x$ in the short axes ($x$) were fitted to be 417.3 µm and 28 µm, respectively, as shown in Fig. 4c. The incident angle $\theta$ was set to 61.5°, which is close to the angle corresponding to the maximum LOF calculated by the 3D-Raytracing method (Supplementary Fig. 10 and Fig. 11). The high extinction ratio (>25 dB) of linear polarization for the laser beam is ensured to avoid the LOF induced by the spin-orbit interaction when the particle is illuminated by a circularly polarized light[36–40] (Supplementary Fig. 12).

Figure 4b shows the snapshots of the lateral movement of an oil droplet with a three-phase contact line radius $r = 9.7$ µm when the polarization angle $\alpha$ is periodically switched between −45° and 45°

(Supplementary Movie 1). To avoid oil droplets escaping from the linear optical trap, the polarization angle is switched circularly with an average period of ~ 63 s. It is seen that the oil droplet moves along the -$y$ direction at $\alpha = −45°$ and then reversely moves along the +$y$ direction when the polarization is switched to 45°, which is consistent with the numerical calculation results of 3D-Raytracing (Supplementary Fig. 10). Using the Hough circle transforms[62] (see Methods) to trace the center of the oil droplet frame by frame, the lateral displacement of the droplet was measured and shown in Fig. 4e. Take the center of the beam as the zero point of lateral displacement, as shown by the gray dotted line. The maximum lateral displacements in +$y$ and -$y$ directions were measured to be 112.9 µm and 111.1 µm, respectively.

The total LOF $F_y^{total}$ in the experiment should consist of the LOF $F_y^{lat}$ and the optical gradient force $F_y^{grad}$. As shown in Fig. 4d, the total LOF $F_y^{total}$ (cyan and blue lines) and gradient force $F_y^{grad}$ (red line) were numerically calculated as a spatial function along the $y$ direction using the 3D-Raytracing method with the experimental parameters. Due to the much longer waist $w_y$ of the laser beam in $y$ axes, the optical gradient force $F_y^{grad}$ is one order of magnitude smaller than the LOF $F_y^{lat}$. As a result, the total LOF $F_y^{total}$ equals almost to the LOF $F_y^{lat}$ without loss of accuracy, and the observed lateral movement of the oil

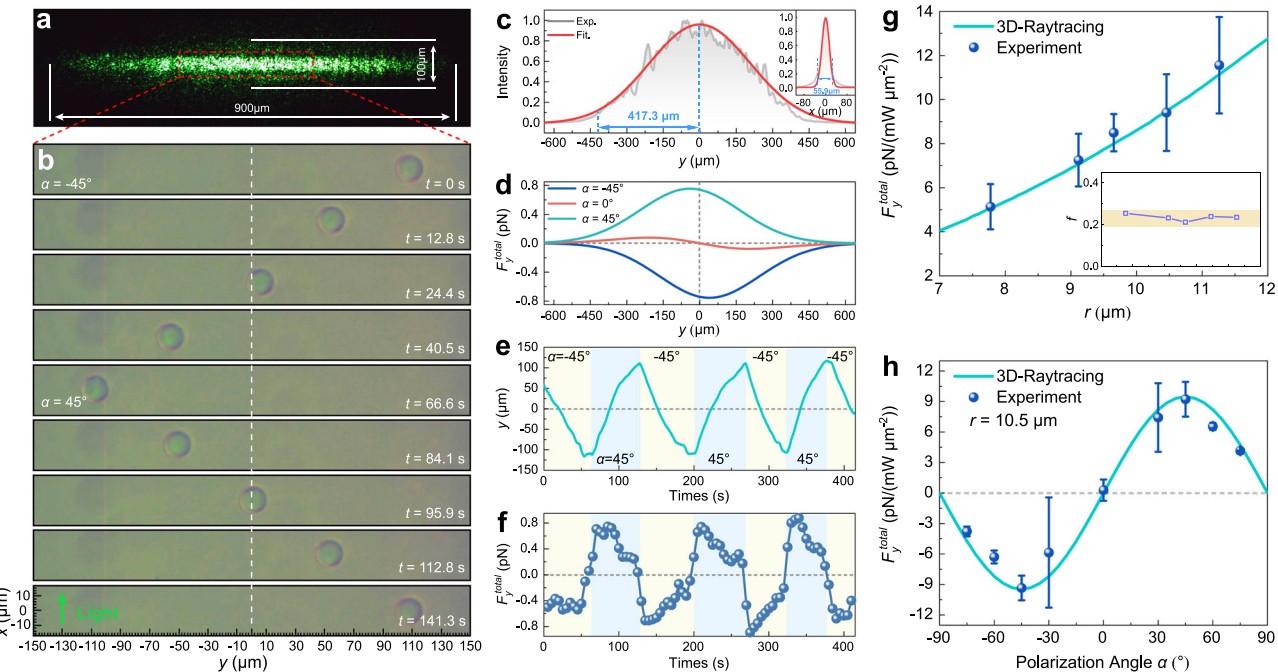

**Fig. 4 | Experimental observation of lateral dynamics of dodecane oil droplets caused by the polarization-induced mirror symmetry breaking. a** Elliptical beam profile observed under 4 × objective at the incident angle of $\theta = 61.5°$. **b** Snapshot of lateral movement of an oil droplet with a radius $r = 9.7\,\mu m$ at the polarization angles $\alpha = \pm45°$ under the 10 × objective (NA = 0.11). The laser power was set to 1.83 W. **c** The Gaussian waist radii along the long ($y$) and short ($x$) axes are estimated as 417.3 $\mu m$ and 28 $\mu m$ (shown in inset), respectively. **d** The total lateral optical force $F_y^{total}$ at the polarization angles $\alpha = -45°$ (blue line), 0° (red line) and 45° (cyan line) were numerically calculated as a spatial function along $y$ direction using the 3D-Raytracing method with the experimental parameters. The total lateral optical force $F_y^{total}$ equals approximately the lateral optical force induced by

the polarization since the gradient force $F_y^{grad}$ is one order of magnitude smaller and could be neglected. **e, f** The variations of the lateral displacement (**e**) and the total lateral optical force $F_y^{total}$ (**f**) of the oil droplet with the polarization angle switching periodically between $\alpha = -45°$ and $\alpha = 45°$. **g** The dependence of the normalized lateral optical force on the radius $r$ of the oil droplets at the polarization angle $\alpha = 45°$. The inset shows the relationship between the drag coefficient $f$ and the radius $r$. **h** The dependence of the normalized lateral optical force on the polarization angle with radius $r = 10.5\,\mu m$. In (**g, h**) the cyan line denotes the lateral optical force calculated by the 3D-Raytracing method, and blue dots denote the lateral optical force measured experimentally. Error bars indicate standard deviation of three independent measurements.

droplet could be considered as a dominant dynamic from the LOF $F_y^{lat}$. Hence, the LOF $F_y^{lat}$ could be measured by using the Stokes formula $F_y^{total} = F_y^{lat} = 6f\pi\eta R_0 V$[63,64] with the viscosity $\eta$ of the water, lateral velocity $V$ of the droplet, radius $R_0$, and dimensionless drag coefficient $f$. $R_0$ is the radius of the sphere corresponding to the spherical cap in water constituting the lenticular droplet. In our experiments, the droplets of different sizes should have the same drag coefficient $f$[63], because $f$ is related to the shape of the droplet, which is uniquely determined by the three-phase contact line radius and the contact angle (Supplementary Note 6). Here, the drag coefficient $f$ is obtained by the LOF calculated by the 3D-Raytracing method (see Methods). For the oil droplet in Fig. 4b, $f$ is fitted to be 0.212, and the corresponding LOF $F_y^{lat}$ ($\sim F_y^{total}$) is obtained and shown in Fig. 4f. The measured average maximum LOF is 0.79 pN (-0.71pN) for $\alpha = 45°$ ($\alpha = -45°$). To determine the average drag coefficient $f_a$, multiple experiments using different radius droplets were performed to obtain the drag coefficient $f$ and the normalized LOF at $\alpha = 45°$, as shown in Fig. 4g. The cyan line denotes the LOF calculated by the 3D-Raytracing and blue dots denote the measured LOF. The drag coefficients $f$ of different radius droplets are shown in the inset. The average drag coefficient $f_a = 0.234$ can be obtained for the floating droplets with a 6.5% relative deviation. The slight relative deviation is greatly consistent with the hydrodynamic prediction that the three-phase droplets have the same drag coefficient, which rigorously proves the validity of LOF measurement.

Furthermore, we investigated the dependence between LOF and polarization angle by rotating the half-wave plate (Supplementary Movie 2). The lateral displacements and speeds of the droplet at different polarization angles are extracted and shown in Supplementary Fig. 13, which confirms that the displacement and speed of

the droplet can be tunable by the polarization and achieve maximum at $\alpha = -45°$ or $\alpha = 45°$ (Supplementary Note 10). The normalized LOF acting on a lenticular oil droplet with three-phase contact line radius $r = 10.5\,\mu m$ at different polarization angles is calculated by 3D-Raytracing (cyan line) and experimentally measured (blue dots) using the average drag coefficient $f_a$, as shown in Fig. 4h. It is seen in Fig. 4h that the LOF $F_y^{lat}$ ($\sim F_y^{total}$) acting on the droplet exhibits obvious sinusoidal polarization dependence, $F_y^{total} \propto \sin(2\alpha)$. This dependence is in good agreement with the dual-dipole theory and 3D-FDTD simulation, ascertaining the LOF origin from the dual-dipole scattering interaction and the polarization-induced MSB. More surprisingly, the maximum LOF efficiency of 9.22 pN/mW/$\mu m^{-2}$ at $\alpha = 45°$ is achieved, almost three orders higher than the recently realized one via spin-orbit interaction[37] and two orders higher than achieved through the inhomogeneity of the spin momentum[42] (Supplementary Note 9). The ultra-high LOF efficiency may originate from three contributions: (1) the mirror asymmetry of the lenticular oil droplet about the $xy$-plane could enhance the MSB of the scattering light about the $xz$-plane, thereby increasing the LOF (Supplementary Fig. 1); (2) the maximum degree of MSB for the 500 nm radius particle is up to 0.2 by the diagonal polarization is 2.8 times higher than that (0.07) by the circularly polarized light at the incident angle of $\theta = 75°$; (3) the air-water interface could enhance the scattering cross-section of the particle. The scattering cross-section of a 500 nm radius particle semi-floating at the interface achieves a maximum of up to 2.7 $\mu m^2$, which is 1.8 times higher than that (1.5 $\mu m^2$) of a particle fully in the air. The high efficiency allows us to achieve a pN LOF at a relatively weak optical intensity (~0.09 mW/ $\mu m^2$), avoiding laser-induced bio-damage, thus breaking the LOF

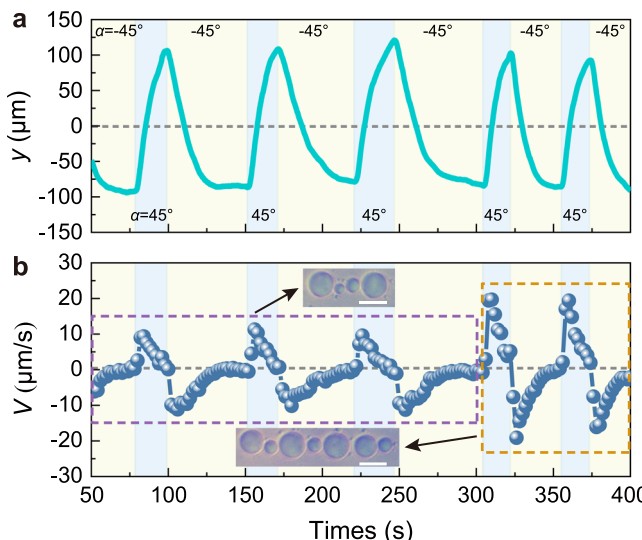

**Fig. 5 | Experimental observation of lateral movement of an aggregate composed of multiple oil droplets. a, b** The variations of the lateral displacement (**a**) and velocity (**b**) of an aggregate composed of multiple oil droplets with the polarization angle switching circularly between $\alpha = -45°$ and $\alpha = 45°$. The insets in (**b**) show snapshots of different numbers of oil droplets in the linear trap at 156 s and 326 s, respectively. The scale bars in the inset equal 30 μm.

constraints and enabling wider potential applications in optical manipulation scenarios.

In the experiment, the lateral movement of an aggregate composed of multiple oil droplets was also observed (Supplementary Movie 3). The lateral displacement and velocity of the aggregate are extracted from the video and shown in Fig. 5. As shown in Fig. 5a, the aggregate exhibits a stable periodic motion when the polarization angle is switched between $\alpha = -45°$ and $\alpha = 45°$, and the maximum displacement is about 100 μm. This indicates that the LOF caused by polarization-induced MSB can not only exist on a single particle but also be applicable to the aggregate composed of multiple particles. Besides, when the number of oil droplets in the linear trap increased from four to seven at 300 s, correspondingly, the movement velocity of the aggregates also increased from 10 μm/s to 20 μm/s, as shown in Fig. 5b. The increased velocity originates from the long-range hydro-dynamic amplification effect[37]. The two insets in Fig. 5b show snapshots of oil droplets in the linear trap at 156 s and 326 s, respectively.

## Discussion

In summary, we report a general polarization-induced MSB mechanism and LOF in an $SO$(2) continuous rotationally symmetric object. The emergence of MSB is solely through the utilization of a simple LP plane wave, breaking the constraints imposed by the chiral material, nanostructure substrate, and complex structured light on LOF. The proposed dual-dipole theoretical model can well describe the MSB mechanism and LOF, revealing the physical origin of the dual-dipole scattering interaction. It is also demonstrated that, for Rayleigh, Mie, and tens-micron-size particles semi-floating at the air-water interface, the MSB and the LOF can exist with a sinusoidal dependence on the polarization angle, providing an efficient way to tune the magnitude and direction of the LOF only using the polarization orientation. Particularly, an ultra-high efficiency of 9.22 pN/mW/μm$^{-2}$ could be implemented experimentally for the LOF to manipulate a tens-of-micron-size particle. The high efficiency and the simple configuration open up extra freedom of optical manipulation for the targeted transport of droplets (drug or surfactant solution) in the fields of medical diagnostics and analytical chemistry[58,59].

## Methods
### Numerical simulations

The 3D-FDTD (Ansys Lumerical) method was performed to numerically simulate the scattering field and calculate optical forces acting on particles, and the 3D-FVEM (COMSOL Multiphysics) was utilized to investigate the far-field radiation. In the simulation, the PS particle ($n_p = 1.5983$) was placed at the interface of air ($n_1 = 1$) and water ($n_2 = 1.337$) under the illumination of an LP plane wave ($\lambda = 532$ nm). The mesh size was set to $\lambda/8/n$[28] for a 1 μm particle in the 3D-FVEM simulation, and the mesh size was set to $R/25$ for particles with a radius $R$ in the 3D-FDTD simulation. In all simulations, perfectly matched layers were used as a boundary condition.

The LOF on a particle was calculated by integrating the Maxwell stress tensor $T$ over a surface surrounding the particle:

$$F_y = \frac{1}{2} \oint_s \sum \text{Re}\langle T \rangle_y ds \tag{7}$$

$$\langle T \rangle = \varepsilon_0 EE^* + \mu_0 HH^* - \frac{1}{2} I^{\leftrightarrow} \left( \varepsilon_0 |E|^2 + \mu_0 |H|^2 \right) \tag{8}$$

where $\langle T \rangle$ is time-averaged Maxwell stress tensor, $I^{\leftrightarrow}$ is the unitary matrix.

### Calculation of the degree of mirror symmetry breaking of lateral scattering

In the 3D-FDTD simulation, the far field profile on a hemispherical surface is 1 meter from the model calculated by the standard 3D far field. The normal component of the Poynting vector can be calculated directly from the far-field scattering electric field by using the plane wave relationships between E and H:

$$P(\phi, \varphi) = n \sqrt{\frac{\varepsilon_0}{\mu_0}} |E_{far}(\phi, \varphi)|^2 \tag{9}$$

where $n$, $\varepsilon_0$ and $\mu_0$ are the background refractive index, vacuum permittivity and permeability, respectively. $\phi$ is the polar angle and $\varphi$ is the azimuth angle. The scattering power can be obtained by integrating the Poynting vector per unit area and be expressed as:

$$W_{sca} = \frac{1}{2} \int \int \text{Re}[P(\phi, \varphi)] \sin \phi d\phi d\varphi \tag{10}$$

The scattering momentum per unit time (in Minkowski formulation) can be expressed as:

$$M_{sca} = \frac{n W_{sca}}{c} \hat{r} \tag{11}$$

where $c$ is the speed of light in vacuum, $\hat{r}$ is the unit vector in the scattering direction.

The degree of mirror symmetry breaking is defined as the ratio of the difference of the scattering momentum along the $+y$ direction and the $-y$ direction to the total scattering momentum along the $y$ direction:

$$D_{MSB} = \frac{M_{+y}^{sca} - M_{-y}^{sca}}{M_{+y}^{sca} + M_{-y}^{sca}} \tag{12}$$

Hence, in combination with Eqs. (9)–(11), $D_{MSB}$ can be obtained only by calculating the far-field scattering electric field component. Finally, the LOF can be expressed as

$$F_y = -\frac{D_{MSB}}{c} \gamma C_{sca}^{total} I_{inc} \tag{13}$$

Here, $C_{sca}^{total} = n_1 C_{sca}^{air} + n_2 C_{sca}^{water}$, $C_{sca}^{air}$ and $C_{sca}^{water}$ are the scattering cross-section of PS particles in air and water, respectively. $n_1$ and $n_2$ are the refractive index of air and water. The scattering cross-section is obtained by the integral of the scattering Poynting vector. It should be noted that if the particle is in a homogeneous medium (i.e. air or water), as shown in Fig. 1, the total scattering cross-section has only one term. $I_{inc}$ is the incident light intensity. $\gamma$ is defined as the ratio of the total scattered momentum in the $y$ direction to the total scattering momentum in all directions

$$\gamma = \frac{M_{+y}^{sca} + M_{-y}^{sca}}{M_{total}^{sca}} \quad (14)$$

### Sample preparation

Micrometer-sized floating liquids were prepared according to the following procedure. Dilute 0.5 μL dodecane solution (Sigma-Aldrich, > 99% purity, refractive index $n = 1.421–1.423$) in 1 mL of $n$-pentane solution (Sigma-Aldrich, anhydrous, >99% purity) in a volume ratio of 1:2000. Sonicate for 10 min to fully dissolve dodecane in $n$-pentane. A single small drop of the solution (~0.5 μL) was rapidly and gently released from a 0.5 μL syringe to the surface of deionized water (Aladdin, W119424-25L). Pentane evaporated rapidly, spontaneously forming dodecane oil droplets (three-phase contact line diameter $d \sim 20\ \mu m$) on the water surface. Since water and oil are immiscible, oil droplets on the water surface can remain stable for a long time.

### Experimental scheme

An LP monochromatic CW laser ($\lambda = 532$ nm, 1.83 W, Changchun New Industries Optoelectronics Tech. Co., Ltd., MGL-V-532) was used to verify the proposed MSB mechanism and LOF. A cylindrical lens (CL) with a focal length of 100 mm and a collimator-beam expander lens set are used to shape the beam, and the beam is focused obliquely on the liquid surface through a mirror to form a linear trap (Supplementary Note 8). The CL could be rotated precisely to make the linear trap along the $y$ direction to ensure experimentally that the oil droplets only move along the $y$ direction instead of the $x$ direction when illuminated by the beam. The set of the collimator-beam expander lens consists of three positive lenses L1, L2 and L3, of which the focal length of L1 is 50 mm, and the focal lengths of L2 and L3 are 250 mm. The half-wave plate (HWP) is rotated to change the linear polarization direction of the beam, but not change the linear shape of the beam. When the relationship between the fast axis angle of the HWP and the polarization angle $\alpha$ after a silver-plated mirror (Thorlabs, PF20-03-P01) is calibrated in our experiments, the linear polarization orientation of the incident light acting on the sample can be determined indirectly by the HWP fast axis angle. To facilitate the observation of the movement of the oil droplets, the sample was prepared in a transparent glass vessel (~5.5 cm × 5.5 cm × 8 mm), and a white LED illuminated the sample from above the liquid surface. A 10 × objective lens with a long working distance (Nikon, NA = 0.11) was placed under the sample for imaging, and a CCD with a frame rate of 16 was connected to the computer to record the movement of the oil droplets in real-time. An edge-pass filter (Thorlabs, FESH0500) was used to remove scattered light so that the movement of the oil droplets could be clearly recorded by the CCD. In addition, the system is placed on an optical shock-absorbing platform to reduce external vibrations. A semi-enclosed transparent box encloses the sample, with holes only on one side of the incident light to avoid airflow-induced movement of oil droplets.

### Particle tracing

Decomposed into a series of frames is the oil drop motion video that the CCD captured. The Hough circle transforms algorithm[62] is used to precisely locate the center of the oil droplet in each frame, and the motion trajectory of the oil droplet can be obtained. The center of the linear trap has been determined before adding the filter and verified experimentally, as shown in Fig. 4a.

### Experimental data analysis

To measure the LOF acting on the oil droplet through the Stokes formula $F_y^{lat} = 6f(h/R_0)\pi\eta R_0 V$, it is necessary to determine the drag coefficient $f$ of the oil droplet. According to the theoretical analysis in Ref. 63 the drag coefficient $f$ depends only on $h/R_0$ accounting for the hydrodynamic particle-interface interaction, where $R_0$ is the radius of the sphere corresponding to the spherical cap in water constituting the lens-shape droplet and $h$ is the height from the top of the sphere in the air to the liquid surface. In the experiment, the shape of the droplet is uniquely determined by the three-phase contact line radius and the contact angle (Supplementary Note 6). Therefore, oil droplets of different sizes should have the same ratio $h/R_0$ and thus the same drag coefficient $f$ in the same experimental environment (same interfacial tension). Here, the drag coefficient $f$ is determined by the LOF calculated by the 3D-Raytracing method. We perform the 3D-Raytracing method to calculate the maximum LOF of the oil droplet under the same experimental parameters and substitute it into the Stokes formula as the force result to obtain $f = F_y^{lat}/(6\pi\eta R_0 V)$. Velocity $V$ corresponds to the maximum motion velocity at $\alpha = \pm 45°$. The $f$ of each oil droplet is the average value obtained by three periodic motions of the oil droplet at $\alpha = \pm 45°$.

## Data availability

The data that support the findings of this study are available from the corresponding author upon request.

## Code availability

The code used in this work is available from the corresponding author upon request.

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

## Acknowledgements

The author would like to acknowledge financial support from National Natural Science Foundation of China (12174155 to Jianhui Yu, 12174156 to Wenguo Zhu, 62105125 to Jieyuan Tang, 61675092 to Jianhui Yu), National Key Research and Development Program of China (2021YFB2800801 to Jianhui Yu), Natural Science Foundation of Guangdong Province for Distinguished Young Scholar (2020B1515020024 to Jianhui Yu), Key-Area Research and Development Program of Guangdong Province (2019B010138004 to Jianhui Yu), Guangdong Basic and Applied Basic Research Foundation (2021A1515110667 to Zhen Che), Special Project in Key Fields of the Higher Education Institutions of Guangdong Province (2020ZDZX3022 to Jianhui Yu) and the Fundamental Research Funds for the Central Universities.

## Author contributions

Yu Zhang, Zhibin Li, Zhen Che, and Jianhui Yu designed the experiments, Yu Zhang, Zhibin Li, and Zhen Che performed the experiments, Yu Zhang, Zhen Che, Wang Zhang, and Yusen Zhang performed the 3D-FDTD numerical simulations, Zhibin Li, Ziqi Lin, and Zhan Lv performed the FVFEM numerical simulations, Chunling Wu, Longwei Han, and Wenguo Zhu performed the 3D-Raytracing numerical calculations, Yu Zhang and Jianhui Yu performed the theoretical derivation, Jieyuan Tang, Wenguo Zhu, Yi Xiao, Huadan Zheng, Yongchun Zhong, Zhe Chen, and Jianhui Yu supervised the project, Yu Zhang, Zhibin Li, and Jianhui Yu wrote the paper, Jianhui Yu conceived the idea, and all authors contributed analysis tools.

## Competing interests

The authors declare no competing interests.
