## [Peer Review File · Nature Communications]

Dynamics of polarization-tuned mirror symmetry breaking in a rotationally symmetric systemREVIEWER COMMENTS

Reviewer #1 (Remarks to the Author):

This work reports on the theoretical and experimental investigation of the lateral optical force caused by mirror-asymmetric (diagonal) linear polarization of paraxial light incident obliquely on a cylindrically-symmetric matter. The paper is basically well written, and it is nice to have an experimental measurement of such lateral force. However, I find the main concept not too surprising or conceptually novel. It is quite obvious that the scattering diagram of a mirror-asymmetric incident field is mirror-asymmetric and hence causes a lateral radiation force. For example, a mirror-asymmetric circular polarization can cause a similar lateral force via asymmetric spin-Hall scattering [37,43].

I did not study the vast literature on such forces, but found an example of the diagonal-polarization-induced lateral force in theoretical paper [43] from 2014. There, the system in Fig. 3(c) with a spherical particle on top of a planar glass-air interface is cylindrically symmetric about the x-axis, and the lateral force F_y appears for diagonal polarizations of paraxial light incident obliquely from the glass side: $\chi = +1, -1$ in Fig. 3(d). Compared to the dual-electric-dipole mechanism considered in the present paper, the lateral force in Ref. [43] has a dual-electric-magnetic-dipole origin and the same polarization-angle dependence. I do not know if precisely this kind of lateral force has been measured experimentally (for circular polarizations it has been [37]), because I am not following this field. Perhaps, experts working in this area could provide more references.

Thus, I think that this work is borderline for Nature Communications. The authors and other expert reviewers should check and revise the novelty statements. I also have a couple of minor remarks:

1. The particles in Fig. 4(b) are almost invisible. There should be a way to increase the contrast or to use additional markers.
2. The radiation forces and torques are not necessarily related to the translational and rotational symmetries of the system, as mentioned in the Introduction (Noether's theorem). Absorption or gain also causes forces and torques on symmetric objects: e.g., torques on absorptive spherical particles.
3. The title sounds too vague and unrelated to the main subject of the paper. The paper is about a lateral optical force but it is not even mentioned in the title.

Reviewer #2 (Remarks to the Author):

The manuscript 'Dynamic of polarization-tuned mirror symmetry breaking in a rotationally symmetric system' by Jianhui Yu, et al., propose a concept of lateral optical force by polarization-induced mirror symmetry breaking in a rotationally symmetry system. The lateral optical force could be tunable and is huger comparing to that of the achiral particles of former results. The lateral optical force is sinusoidally dependent on the polarization angle, which is demonstrated by the experiment. The concept is new, the writing is methodical. and the work is systematic. However, I still have several main concerns on the current manuscript:

1. At first glance, only rotating the polarization angle or breaking the mirror symmetry of nano particle cannot induce to the lateral optical force. Actually, we think the concept of lateral optical force proposed here is similar to that acted on the Janus particles such as, Ilic et al., Sci. Adv. 2017;3: e1602738, which also has the mirror symmetry broken. This makes us concern about the novelty of the concept.

2. The authors introduce their works from the perspective of symmetry with the Noether's first theorem, rotational symmetry system and mirror symmetric broken, etc. Also, the authors also use the dipole model to demonstrate the lateral optical force. However, what is the intrinsic direct connection among the symmetry, conservation laws and lateral optical force? The authors should explain. This may be resolved by analyzing the energy-momentum tensor.
3. Actually, there are two types of mirror symmetry broken in the manuscript. One is arisen from the rotation of polarization angle (with respect to the yz-plane), and the other is the mirror symmetry broken of nanoparticles, by the geometry or the refractive index. What are the roles of these two types of mirror symmetry broken in lateral optical force?
4. One of our concerns is the application. The lateral optical force of chiral particles can be applied in chiral sorting and the lateral optical of achiral particles can be applied in the field of biochemistry, etc. However, although the concept of current proposal is novel, the selectivity of particle with specified mirror symmetry broken will provide barriers for future applications.

Some minor revisions:

1. There are too many abbreviations in the manuscript.
2. In line 98, the authors mention that 'It is obviously seen that the polarization can induce the MSB of the scattering field. Accordingly, the asymmetrical light scattering breaks the lateral conservation of optical momentum in the y direction, giving rise to a recoil LOF F_y acting on the particle.' This is less rigorous. Why use the recoil here? Only describe the force is reversal?
3. What is ratio of longitudinal optical force and lateral optical force in these instances?

Reviewer #3 (Remarks to the Author):

The authors present a study that a force occurs in a rotationally symmetric system stemming from the break of the particle polarization symmetry. This work is somehow interesting and delivers some experimental demonstrations. After reading the whole manuscript, I cannot recommend its publication in its current form. Some concerns are as follows,

There lacks a connection from the geometric asymmetry to the polarization asymmetry in Fig. 1. Thus, it may confuse audiences what is the true topic of this work.

The dipole theory is presented in this work to support their conclusions. However, it does not explain why the size cause the force to reverse sign in Fig. 3. This should be classified clearly.

A video showing the successively tuning of the polarization and the particle movement should be given.

The authors simulate PS particles, while use the droplet in the experiment, which is not consistent. Why not using PS particles in the experiment or using droplet in the simulation?

The feasibility of Maxwell stress tensor used in an interface system is not classified.

The manuscript needs an extensive revision. It seems that the writing is quite rush. Lots of English writing problems exhibit inside.

In light of those issues, I am afraid that I cannot recommend its publication in its current form. It may be considered by a specific optics journal after a thorough revision.

REVIEWER COMMENTS

Reviewer #1 (Remarks to the Author):

This work reports on the theoretical and experimental investigation of the lateral optical force caused by mirror-asymmetric (diagonal) linear polarization of paraxial light incident obliquely on a cylindrically-symmetric matter. The paper is basically well written, and it is nice to have an experimental measurement of such lateral force. However, I find the main concept not too surprising or conceptually novel. It is quite obvious that the scattering diagram of a mirror-asymmetric incident field is mirror-asymmetric and hence causes a lateral radiation force. For example, a mirror-asymmetric circular polarization can cause a similar lateral force via asymmetric spin-Hall scattering [37,43].

I did not study the vast literature on such forces, but found an example of the diagonal-polarization-induced lateral force in theoretical paper [43] from 2014. There, the system in Fig. 3(c) with a spherical particle on top of a planar glass-air interface is cylindrically symmetric about the x-axis, and the lateral force F_y appears for diagonal polarizations of paraxial light incident obliquely from the glass side: $\chi = +1, -1$ in Fig. 3(d). Compared to the dual-electric-dipole mechanism considered in the present paper, the lateral force in Ref. [43] has a dual-electric-magnetic-dipole origin and the same polarization-angle dependence. I do not know if precisely this kind of lateral force has been measured experimentally (for circular polarizations it has been [37]), because I am not following this field. Perhaps, experts working in this area could provide more references.

Thus, I think that this work is borderline for Nature Communications. The authors and other expert reviewers should check and revise the novelty statements. I also have a couple of minor remarks:

1. The particles in Fig. 4(b) are almost invisible. There should be a way to increase the contrast or to use additional markers.
2. The radiation forces and torques are not necessarily related to the translational and rotational symmetries of the system, as mentioned in the Introduction (Noether's theorem). Absorption or gain also causes forces and torques on symmetric objects: e.g., torques on absorptive spherical particles.
3. The title sounds too vague and unrelated to the main subject of the paper. The paper is about a lateral optical force but it is not even mentioned in the title.

Reviewer #2 (Remarks to the Author):

The manuscript 'Dynamic of polarization-tuned mirror symmetry breaking in a rotationally symmetric system' by Jianhui Yu, et al., propose a concept of lateral optical force by polarization-induced mirror symmetry breaking in a rotationally symmetry system. The lateral optical force could be tunable and is huger comparing to that of the achiral particles of former results. The lateral optical force is sinusoidally dependent on the polarization angle, which is demonstrated by the experiment. The concept is new, the writing is methodical. and the work is systematic. However, I still have several main concerns on the current manuscript:

1. At first glance, only rotating the polarization angle or breaking the mirror symmetry of nano particle cannot induce to the lateral optical force. Actually, we think the concept of lateral optical force proposed here is similar to that acted on the Janus particles such as, Ilic et al., Sci. Adv. 2017;3: e1602738, which also has the mirror symmetry broken. This makes us concern about the novelty of the concept.
2. The authors introduce their works from the perspective of symmetry with the Noether's first theorem, rotational symmetry system and mirror symmetric broken, etc. Also, the authors also use the dipole model to demonstrate the lateral optical force. However, what is the intrinsic direct connection among the symmetry, conservation laws and lateral optical force? The authors should explain. This may be resolved by analyzing the energy-momentum tensor.
3. Actually, there are two types of mirror symmetry broken in the manuscript. One is arisen from the rotation of polarization angle (with respect to the yz-plane), and the other is the mirror symmetry broken of nanoparticles, by the geometry or the refractive index. What are the roles of these two types of mirror symmetry broken in lateral optical force?
4. One of our concerns is the application. The lateral optical force of chiral particles can be applied in chiral sorting and the lateral optical of achiral particles can be applied in the field of biochemistry, etc. However, although the concept of current proposal is novel, the selectivity of particle with specified mirror symmetry broken will provide barriers for future applications.

Some minor revisions:

1. There are too many abbreviations in the manuscript.
2. In line 98, the authors mention that 'It is obviously seen that the polarization can induce the MSB of the scattering field. Accordingly, the asymmetrical light scattering breaks the lateral conservation of optical momentum in the y direction, giving rise to a recoil LOF F_y acting on the particle.' This is less rigorous. Why use the recoil here? Only describe the force is reversal?
3. What is ratio of longitudinal optical force and lateral optical force in these instances?

Reviewer #3 (Remarks to the Author):

The authors present a study that a force occurs in a rotationally symmetric system stemming from the break of the particle polarization symmetry. This work is somehow interesting and delivers some experimental demonstrations. After reading the whole manuscript, I cannot recommend its publication in its current form. Some concerns are as follows,

1. There lacks a connection from the geometric asymmetry to the polarization asymmetry in Fig. 1. Thus, it may confuse audiences what is the true topic of this work.
2. The dipole theory is presented in this work to support their conclusions. However, it does not explain why the size cause the force to reverse sign in Fig. 3. This should be classified clearly.
3. A video showing the successively tuning of the polarization and the particle movement should be given.
4. The authors simulate PS particles, while use the droplet in the experiment, which is not consistent. Why not using PS particles in the experiment or using droplet in the simulation?
5. The feasibility of Maxwell stress tensor used in an interface system is not classified.
6. The manuscript needs an extensive revision. It seems that the writing is quite rush. Lots of English writing problems exhibit inside.
7. In light of those issues, I am afraid that I cannot recommend its publication in its current form. It may be considered by a specific optics journal after a thorough revision.

-----Comments and Responds-----

Reviewer #1:

Comments to the Author:

This work reports on the theoretical and experimental investigation of the lateral optical force caused by mirror-asymmetric (diagonal) linear polarization of paraxial light incident obliquely on a cylindrically-symmetric matter. The paper is basically well written, and it is nice to have an experimental measurement of such lateral force.

Comment 1: However, I find the main concept not too surprising or conceptually novel. It is quite obvious that the scattering diagram of a mirror-asymmetric incident field is mirror-asymmetric and hence causes a lateral radiation force. For example, a mirror-asymmetric circular polarization can cause a similar lateral force via asymmetric spin-Hall scattering [37,43].

Response: Thank you for your critical and constructive comment. Although it is obvious that a mirror-asymmetric incident field will cause mirror-asymmetric scattering, it does not mean that only the incident field is considered and the mirror symmetry breaking (MSB) is relative to the object with which the light interacts. To the best of our knowledge, it is the first time to propose and verify the mechanism of mirror asymmetry in the rotational symmetry $SO(2)$ system caused by a linear polarization of the incident field. Due to its universality and simplicity, such a mirror-symmetry breaking mechanism will play an essentially important role in various applications using optical force, and become a non-negligible optical force effect.

The literature [37,43] you mentioned presents two novel mechanisms to break the mirror symmetry, but they are very different from the mechanism we proposed here. Literature [37] proposed the optical spin-orbital interaction (SOI) near the interface to break the mirror symmetry and generate a lateral optical force (LOF). However, despite the intrinsic mirror asymmetry of the incident field in a circular polarization state, the mirror symmetry of the SOI-caused spiral light scattering momentum is still unable to be broken without the interface of two mediums. Thus, the interface is indispensable

for the mirror-symmetry breaking in the SOI mechanism, and our proposed MSB mechanism does not require an interface, as shown in Fig. 1 in the main text. Additionally, in the experiment configuration of literature [37], i.e. the cases of the particle immersed fully in the water or fully in the air, the linear polarization incident field with the mirror-asymmetry hardly leads to mirror-asymmetry scattering and LOF. We performed a numerical simulation of these cases through the three-dimensional finite difference time domain (3D-FDTD) method, as shown in Fig. A1. Figure A1 shows the LOFs for a polystyrene (PS) particle ($n_p = 1.5983$ @ $\lambda = 532\text{nm}$) completely immersed in the air, in the water and semi-floating at the air-water interface, respectively. The particle radius is $0.5\mu\text{m}$, and when the particles are completely immersed in air or water, the distance from the particle to the interface is $0.5\mu\text{m}$. Firstly, it is seen that in all three cases, the LOF varies sinusoidally with polarization angle α , i.e., LOF is proportional to $\sin(2\alpha)$. Secondly, when a particle is semi-floating at the air-water interface, the maximum LOF is $0.073\text{pN}/(\text{mW}\mu\text{m}^{-2})$, which is 28 times higher than that when the particle is completely in the air and 7.3 times higher than that when the particle is completely in the water.

In the literature [43], K. Y. Bliokh, et al. present extraordinary transverse momentum of spin and Poynting in the evanescent field (a kind of inhomogeneous field). The transverse momentum exists only in higher-order interaction, such as quadratic dipole-dipole interaction. Such an LOF mechanism requires two necessary conditions: (1) a spatially inhomogeneous optical field; (2) a large size of a particle that is beyond the Rayleigh region. In contrast, our proposed LOF mechanism comes directly from the interaction between a homogeneous field (plane wave) and the rotation symmetry object (arbitrary size particle) and thus gets rid of the limits from the two conditions.

As far as we know, there is still no literature that reports the mechanism to generate a LOF only by a linearly polarized plane wave in rotationally symmetric objects. The LOF mechanism comes simply from the asymmetrical field-object interaction. However, the simplicity of the mechanism yields a large LOF efficiency comparable with the optical pressure force, making it non-negligible in various applications of optical force.

Figure A1. The LOF as a function of polarization angle for particles completely in the air, water and semi-floating at the air-water interface.

Comment 2: I did not study the vast literature on such forces, but found an example of the diagonal-polarization-induced lateral force in theoretical paper [43] from 2014. There, the system in Fig. 3(c) with a spherical particle on top of a planar glass-air interface is cylindrically symmetric about the x-axis, and the lateral force F_y appears for diagonal polarizations of paraxial light incident obliquely from the glass side: $\chi = +1, -1$ in Fig. 3(d). Compared to the dual-electric-dipole mechanism considered in the present paper, the lateral force in Ref. [43] has a dual-electric-magnetic-dipole origin and the same polarization-angle dependence. I do not know if precisely this kind of lateral force has been measured experimentally (for circular polarizations it has been [37]), because I am not following this field. Perhaps, experts working in this area could provide more references.

Thus, I think that this work is borderline for Nature Communications. The authors and other expert reviewers should check and revise the novelty statements.

Response: Thank you for your constructive comment. In literature [43], the article theoretically revealed that the evanescent field carries helicity-independent transverse spin, helicity-dependent transverse Belinfante's spin momentum, and diagonal-polarization-dependent 'imaginary' transverse Poynting vector. Numerical experiments show that the transverse helicity-independent spin can be detected straightforwardly *via* the transverse torque T_y exerted on an absorbing small particle. The helicity-dependent

transverse Belinfante's spin momentum p_y^s and diagonal-polarization-dependent 'imaginary' transverse Poynting vector $\text{Im}\tilde{p}_y$ can be detected *via* a transverse force F_y from the interaction with the probe gold particle. Here, the transverse force F_y consists of two contributions ($F_y \propto \text{Re}(\alpha_e \alpha_m^*) p_y^s + \text{Im}(\alpha_e \alpha_m^*) \text{Im}\tilde{p}_y$), where the transverse diagonal-polarization-dependent force is associated with the transverse imaginary Poynting vector $\text{Im}\tilde{p}_y$ and the maximum force emerges at linearly diagonally polarized of $\pm 45^\circ$. The LOF caused by the transverse Poynting vector in the literature [43] requires two necessary conditions: (1) a spatially inhomogeneous optical field, such as an evanescent field, and (2) a large size of a particle that is beyond the Rayleigh region with higher-order interaction.

In contrast, the LOF mechanism proposed in the present work is novel and very different from that in literature [43], which requires no interface between two mediums and is not subject to the two conditions mentioned above. The proposed LOF mechanism in our work only requires two conditions: (1) the diagonal polarization of the incident field; (2) the object with an $SO(2)$ rotationally symmetry that only one rotation symmetric axis exists. Accordingly, the LOF proposed in our work can be generated in an optical plane wave with a homogeneous field and in an arbitrary size particle.

In our work, two specific instances, a dual-dipole and a semi-floating particle, are investigated theoretically and experimentally for the $SO(2)$ system to confirm the mirror-symmetry breaking mechanism by the diagonally polarized plane wave. The dual-dipole model considers the refraction at the interface, and the LOF originates from the asymmetric mutual scattering between two dipoles located in different media. It should be emphasized that even when the two dipoles with the same electric polarizability are located in a homogeneous medium, the LOF can also exist without the medium interface. Therefore, the mechanism proposed in our work is completely different from the mechanism in the literature [43]. Our work does not require an evanescent wave (inhomogeneous wave) and interface of two mediums, and thus

potentially has a more general application.

Comment 3: The particles in Fig. 4(b) are almost invisible. There should be a way to increase the contrast or to use additional markers.

Response: Thank you for your constructive suggestions. We have revised Fig.4 (b) and increased the contrast of the image to more clearly show the location of oil droplets.

Comment 4: The radiation forces and torques are not necessarily related to the translational and rotational symmetries of the system, as mentioned in the Introduction (Noether's theorem). Absorption or gain also causes forces and torques on symmetric objects: e.g., torques on absorptive spherical particles.

Response: Thank you for pointing out the less rigorous expression in the manuscript. In fact, what we mean is in a non-absorptive system, the breaking of the translational and rotational symmetry of the system causes radiation forces and torques.

Accordingly, we have revised the statement in the introduction to make it more rigorous, and new statements have been added to line 17, page 2.

“For instance, in a non-absorptive system, the translational symmetry breaking of the light-matter system through the exchange of the linear optical momentum gives rise to a push or pull optical force³⁻⁹”

Comment 5: The title sounds too vague and unrelated to the main subject of the paper. The paper is about a lateral optical force but it is not even mentioned in the title.

Response: Thank you for your constructive comments. In our work, the dynamic movement of the semi-floating particles in our experiment is caused by the polarization-induced mirror-symmetry breaking and the LOF. Here, the LOF mechanism (the polarization-tuned mirror-symmetry breaking) is investigated mainly in the present work because it is the physical origin of the transverse dynamics of the semi-floating particles. However, the dynamic transverse movement of the particle closely related to the LOF is the only way to measure such LOF, and the words “lateral optical force” cannot include the mean of the dynamic transverse movement of the particles. Therefore, we think the word “dynamics” in the title is more rigorous than the words

“lateral optical force”.

Accordingly, we revised the “dynamic” in the title as “dynamics” and mentioned the lateral dynamics of the particle movement in the text.

Reviewer #2:

Comments to the Author:

The manuscript ‘Dynamic of polarization-tuned mirror symmetry breaking in a rotationally symmetric system’ by Jianhui Yu, et al., propose a concept of lateral optical force by polarization-induced mirror symmetry breaking in a rotationally symmetry system. The lateral optical force could be tunable and is huger comparing to that of the achiral particles of former results. The lateral optical force is sinusoidally dependent on the polarization angle, which is demonstrated by the experiment. The concept is new, the writing is methodical. and the work is systematic. However, I still have several main concerns on the current manuscript:

Comment 1: At first glance, only rotating the polarization angle or breaking the mirror symmetry of nano particle cannot induce to the lateral optical force. Actually, we think the concept of lateral optical force proposed here is similar to that acted on the Janus particles such as, Ilic et al., Sci. Adv. 2017;3: e1602738, which also has the mirror symmetry broken. This makes us concern about the novelty of the concept.

Response: Thank you for your constructive comment. In the literature you mentioned, firstly, the literature did not discuss and study the LOF mechanism but focused on the stability of torque in phase space. Secondly, the Janus particle is only a special case of the $SO(2)$ rotational symmetry system we proposed, and the physical mechanism of linear polarization-induced MSB has not been proposed in the literature. To the best knowledge of us, it is the first time to propose the LOF mechanism, where only diagonal polarization is needed to break the mirror symmetry. In addition, half of the Janus particles are covered with a gold film, which is absorptive and essentially different from the non-absorptive particles demonstrated in our experiments. Since the gold film on the particle absorbs light, the symmetry breaking in the literature is most likely due to

absorption. Nevertheless, the MSB mechanism and LOF we proposed here are applicable to the $SO(2)$ rotational symmetric system without absorption and with only one rotational symmetry axis. Furthermore, the mechanism proposed here also provides a novel LOF tunability by rotating the linear polarization and high LOF efficiency. As for us, the mechanism is proposed for the first time and is physically novel compared to the literature [Sci. Adv. 2017;3: e1602738] you mentioned. Because the mechanism has a wide generality, we think the proposed mechanism should be widely attractive to readers in a similar field and unignored in various applications of optical manipulations.

Comment 2: The authors introduce their works from the perspective of symmetry with the Noether's first theorem, rotational symmetry system and mirror symmetric broken, etc. Also, the authors also use the dipole model to demonstrate the lateral optical force. However, what is the intrinsic direct connection among the symmetry, conservation laws and lateral optical force? The authors should explain. This may be resolved by analyzing the energy-momentum tensor.

Response: Thank you for your constructive suggestions. We have presented the intrinsic relevance among symmetry, conservation law, and LOF, as shown in Fig. 2 in the main body. When the polarization of the incident light deviates from s - or p -polarization, the mirror symmetry of the scattered field is broken, as shown in Figs. 2b and 2d. To maintain the conservation of transverse momentum, the particle will be subjected to the action of LOF. To relate LOF to the symmetry and conservation laws, we define the degree of mirror symmetry breaking D_{MSB} in terms of the scattering momentum. The degree of mirror symmetry breaking D_{MSB} is defined as the ratio of the difference of the scattering momentum along the $\pm y$ direction to the total scattering momentum along the y direction

$$D_{MSB} = \frac{M_{+y}^{sca} - M_{-y}^{sca}}{M_{+y}^{sca} + M_{-y}^{sca}} \quad (A.1)$$

In equation (A.1), the scattering momentum $M_{\pm y}^{sca} = \frac{nW_{\pm y}^{sca}}{c}$ is obtained through scattering energy W^{sca} . Furthermore, according to the relationship between momentum and force,

we obtain the expression of the LOF

$$F_y = -\frac{D_{MSB}}{c} \gamma C_{sca}^{total} I_{inc} \quad (A.2)$$

Here, γ is defined as the ratio of the total scattered momentum in the y direction to total scattering momentum in all directions. $C_{sca}^{total} = n_1 C_{sca}^{air} + n_2 C_{sca}^{water}$, C_{sca}^{air} and C_{sca}^{water} are the scattering cross-section of PS particle in air and water, respectively. n_1 and n_2 are the refractive index of air and water. I_{inc} is the incident light intensity. It is seen from equation (A.2) that LOF is proportional to D_{MSB} . The dependence of D_{MSB} on polarization angle α and incident angle θ_I is shown in Fig. 2h. Therefore, equation (A.2) connects symmetry, conservation law and LOF.

In a rigorous way, the Lagrange action of the optical field, particle, and their interaction should be carefully constructed to derive the LOF through the MSB as the optical transverse momentum is transferred to the particle. We think it is better to discuss and reveal the relevance between mirror symmetry, rotational symmetry and LOF in other work.

Comment 3: Actually, there are two types of mirror symmetry broken in the manuscript. One is arisen from the rotation of polarization angle (with respect to the yz -plane), and the other is the mirror symmetry broken of nanoparticles, by the geometry or the refractive index. What are the roles of these two types of mirror symmetry broken in lateral optical force?

Response: Thank you for your comment. As you mentioned, the demonstrated light-matter system contains two kinds of MSB, one induced by the rotation of the linear polarization of the incident light (the MSB with respect to the xz -plane), and the other arises from the particle geometry or refractive index (the MSB with respect to the xy -plane).

It should be noted that the MSB mentioned in our work mainly refers to the MSB of the scattering field respective to the xz -plane, where the particle is mirror symmetric about the xz -plane. Therefore, the LOF caused by the MSB is along the y -axis. To the best of our knowledge, the MSB mechanism is first proposed and it only needs two

conditions: (1) the object has $SO(2)$ rotational symmetry, that is, the object is mirror symmetric about the xz -plane, (2) the oblique incidence of a diagonally polarized plane wave. To verify the MSB mechanism, six objects with different geometries with $SO(2)$ rotational symmetry in Fig. 1a and Fig. 2a are demonstrated to break mirror symmetry only by diagonal polarization. Here, these objects are classified into two types: (1) the object that is mirror-symmetric about the xy -plane, such as a cylinder; (2) the object that is mirror-asymmetric about the xy -plane, such as a hemisphere, a cone, or a sphere semi-floating at the air-water interface. For the object having mirror symmetry with respect to the xy -plane, an oblique incidence ($0^\circ < \theta_I < 90^\circ$) becomes an essential condition to break the xy -plane mirror symmetry and generate the LOF by the diagonally polarized plane wave, such as a cylinder as shown in Figs. A2a and A2e. When the incidence of light becomes vertical ($\theta_I = 90^\circ$), the mirror symmetry cannot be broken by the diagonal polarization and the LOF disappears for the object with a xy -plane mirror symmetry, as shown in Figs. A2b and A2f. However, for the object with xy -plane mirror asymmetry (such as a hemisphere), the vertical incidence of light can also break the xz -plane mirror symmetry by the diagonal polarization and generate the LOF along the y direction, as shown in Figs. A2d and A2f. In particular, when the incident light is vertical ($\theta_I = 90^\circ$), the maximum LOF efficiency of the hemisphere is 2.2 times higher than that for the oblique incidence ($\theta_I = 60^\circ$). This indicates that, for the object with the xy -plane mirror asymmetry, the degree of MSB induced by the vertical illumination is higher than that for the oblique illumination.

In summary, for an $SO(2)$ rotationally symmetric object with the xy -plane mirror symmetry, the oblique incident of light is an essential condition to break the xy -plane mirror symmetry of the light-matter system. For an object with mirror asymmetry about the xy -plane, the diagonal polarization of the incident light (even at vertical incidence) could break the xz -plane mirror symmetry and generate the LOF along the y direction. Therefore, the breaking of xy -plane mirror symmetry becomes one of the essential conditions to generate the LOF.

Figure A2. (a) For an $SO(2)$ rotationally symmetric object with mirror symmetric about the xy -plane, the mirror symmetry of the light-matter interaction about the xy -plane is broken by an obliquely incident ($0^\circ < \theta_i < 90^\circ$) linearly polarized plane wave. The LOF can be generated by the diagonally polarized light. (b) For an $SO(2)$ rotationally symmetric object that is mirror symmetric about the xy -plane, the mirror symmetry of the light-matter interaction about the xy -plane is maintained when the incident angle is 90° . The LOF disappears regardless of polarization orientation. (c-d) For an $SO(2)$ rotationally symmetric object with mirror asymmetry about the xy -plane, when the incident light is incident obliquely ($0^\circ < \theta_i < 90^\circ$) or grazingly ($\theta_i = 90^\circ$), LOF can be induced when the polarization angle deviates from s - or p -polarization. (e-f) The dependence of the LOF of the cylinder and hemisphere on polarization angle when the incident angle is 60° and 90° , respectively.

Comment 4: One of our concerns is the application. The lateral optical force of chiral particles can be applied in chiral sorting and the lateral optical of achiral particles can be applied in the field of biochemistry, etc. However, although the concept of current proposal is novel, the selectivity of particle with specified mirror symmetry broken will provide barriers for future applications.

Response: Thank you for your concern. The proposed novel LOF is essentially different from the chirality-related LOF. As reviewer's concern, the LOF can be used for chiral sorting. For example, in literature [28], chiral sorting of semi-floating chiral liquid crystals at the gas-liquid interface can be achieved using s - or p -polarized linearly polarized plane waves. However, in our proposed LOF generation mechanism, obliquely incident s - or p -polarized linearly polarized plane waves will not generate LOF on $SO(2)$ rotationally symmetric objects. Therefore, when sorting mirror-

symmetry breaking particles (chiral particles), *s*- or *p*-polarized light can be chosen to avoid the polarization-induced MSB and implement the chiral sorting caused by the novel LOF proposed in the literature [28]. Therefore, our proposed polarization-induced MSB will not become a barrier to the chiral sorting application. Furthermore, our proposed MSB could not only provide an extra tunable dimension for the LOF but potentially provide an application in size or geometry sorting.

Comment 5: There are too many abbreviations in the manuscript.

Response: Thank you for your constructive suggestions. We have revised the main text and removed infrequently cited abbreviations such as spin-orbit interaction (SOI), the degree of mirror symmetry breaking (DMSB) and dual-dipole separation (DDS).

Comment 6: In line 98, the authors mention that ‘It is obviously seen that the polarization can induce the MSB of the scattering field. Accordingly, the asymmetrical light scattering breaks the lateral conservation of optical momentum in the *y* direction, giving rise to a recoil LOF F_y acting on the particle.’ This is less rigorous. Why use the recoil here? Only describe the force is reversal?

Response: Thank you for your comment. Conservation of momentum means that the particles must acquire equal and opposite mechanical momentum through asymmetric light scattering to cancel out the total transverse momentum. Figure A3 shows the LOF caused by a non-symmetric scattering. When the polarization angle $\alpha = -45^\circ$ (45°), the scattering field in the $+y$ ($-y$) direction is larger than that in the $-y$ ($+y$) direction. Therefore, lateral momentum conservation gives rise to a LOF along the $-y$ ($+y$) direction exerting on the particle. Meanwhile, in our work, we also gave the expression of LOF related to momentum:

$$F_y = -\frac{D_{MSB}}{c} \gamma C_{sca}^{total} I_{inc} \quad (A.3)$$

Here, the degree of mirror symmetry breaking D_{MSB} is defined as the ratio of the difference of the scattering momentum along the $+y$ and $-y$ direction to the total scattering momentum along the y direction, that is, $D_{MSB} = (M_{+y}^{sca} - M_{-y}^{sca}) / (M_{+y}^{sca} + M_{-y}^{sca})$. The

scattering momentum changes with the polarization angle of the incident light. Additionally, the term recoil was used in the original manuscript to describe that the force was caused by changes in lateral scattering momentum. In order to express it rigorously, we use “lateral optical force” or “LOF” uniformly in the manuscript to describe the optical force induced by the mirror symmetry breaking.

Figure A3. Polarization-dependent LOF.

Comment 7: What is ratio of longitudinal optical force and lateral optical force in these instances?

Response: Thank you for your comment. In the proposed mechanism, the longitudinal optical force is the scattering force along the propagation direction of the incident light. We calculated the resultant force of F_x and F_z in the incident plane and projected it into the incident direction to obtain the longitudinal optical force F_s , as shown in Fig. A4. Figures A4a and A4b show the longitudinal optical force F_s and LOF F_y on PS particles with radii of 50nm and 500nm at $\alpha = 45^\circ$, respectively. The ratio of the longitudinal optical force to the LOF is shown in Figs. A4c and A4d. It is seen that the magnitude of the LOF is comparable to the longitudinal scattering force and the ratio between the two forces is related to the particle size and incident angle. Except for certain incident angles, the ratio between the two optical forces is always less than 10. Therefore, the LOF and the longitudinal scattering force are of the same order of magnitude in our proposed mechanism.

Figure A4. (a) The dependence of the normalized longitudinal optical force and the LOF of the PS particle with a radius of 50nm on the incident angle. (b) The dependence of the normalized longitudinal optical and the LOF of the PS particle with a radius of 500nm on the incident angle. (c) The ratio of the longitudinal optical force and LOF of PS particles with a radius of 50nm. (d) The ratio of the longitudinal optical force and LOF of PS particles with a radius of 500nm.

Reviewer #3:

Comments to the Author:

The authors present a study that a force occurs in a rotationally symmetric system stemming from the break of the particle polarization symmetry. This work is somehow interesting and delivers some experimental demonstrations. After reading the whole manuscript, I cannot recommend its publication in its current form. Some concerns are as follows:

Comment 1: There lacks a connection from the geometric asymmetry to the polarization asymmetry in Fig. 1. Thus, it may confuse audiences what is the true topic of this work.

Response: Thank you for your constructive comment. We have modified Fig. 1 to more clearly show the connection between geometric asymmetry and polarization asymmetry.

Comment 2: The dipole theory is presented in this work to support their conclusions. However, it does not explain why the size cause the force to reverse sign in Fig. 3. This should be classified clearly.

Response: Thank you for your constructive comment. The reason why the dual-dipole model can support our conclusions is that the established dual-dipole model has only one rotational symmetry axis and can be regarded as a special case of the $SO(2)$ rotational symmetry system shown in Fig. 1. In our work, the semi-floating sphere can be considered as two different scatters. For simplicity, the two dipoles could be well used to describe the two scatters as a lowest-order approximation (Discrete Dipole Approximation). Consequently, the distance between the dipoles is closely related to the particle size (particle radius R). Here, the dependence of LOF on particle radius R can be calculated through numerical simulation, as shown in Fig. 3d. The particle radius ranges from 300 nm to 1000 nm and equivalently induces the phase retardation between the dipoles making the dipole moment term reverses as discussed in following. It is seen that small incident angles can easily induce a reversal of the LOF, as shown in the inset. This oscillation of the LOF with particle size at the small incident angle originates from two contributions.

On the main contribution, in the dual-dipole theory, the LOF is related to the dipole moment term (See equation 1 in main text). The dipole moment term can be expressed as

$$\text{Re}(d_{2y}d_{1z}^*) = \frac{1}{2} \alpha_{e,1} \alpha_{e,2} |E_{inc}|^2 \sin(2\alpha) \psi_1(\theta_I, z_1) \quad (\text{A.4})$$

$$\text{Re}(d_{1y}^*d_{2z}) = \frac{1}{2} \alpha_{e,1} \alpha_{e,2} |E_{inc}|^2 \sqrt{\varepsilon_{12}} \sin(2\alpha) \psi_2(\theta_I, z_1) \quad (\text{A.5})$$

Here $\alpha_{e,i}$ is the dipole electric polarizability, which is related to the particle radius R and can be expressed by the Clausius-Mossotti relationship $\alpha_{e,i} = 4\pi\varepsilon_i R^3 [(\varepsilon_p - \varepsilon_i) / (\varepsilon_p + 2\varepsilon_i)]$.

The $\psi_1(\theta_I, z_1)$ is closely related to the cosine function of the phase retardation δ_{d12} ,

which can be expressed as

$$\psi_1(\theta_l, z_1) = t_s(\theta_l) \sin(\theta_l) \left[\cos(\delta_{d12}) + r_p(\theta_l) \cos(\delta_{ref} - \delta_{d12}) \right] \quad (\text{A.6})$$

$$\psi_2(\theta_l, z_1) = t_p(\theta_l) \sin(\theta_l) \left[\cos(\delta_{d12}) + r_s(\theta_l) \cos(\delta_{ref} - \delta_{d12}) \right] \quad (\text{A.7})$$

, where $\delta_{d12} = k_2 \left[z_1 \sqrt{\epsilon_{12}} \cos(\theta_l) + z_2 \sqrt{1 - \epsilon_{12} \sin^2(\theta_l)} \right]$ is proportional to the product of the dual-dipole separation z_i (closely related to particle radius R) and the cosine function of incident angle θ_l . In the dual-dipole theory, a spherical PS particle semi-floating at the gas-liquid interface is equivalent to two dipoles of the same size. The distance between the dipole from the interface is considered equal to the radius of the PS particle, $z_1 = z_2 = R$. Therefore, according to equations (A.6) and (A.7), the magnitude of the ψ depends on size-dependent phase retardation δ_{d12} and ψ will oscillate as the particle size R increases. The oscillatory dependence of the ψ in equations (A.6) and (A.7) on particle size R is calculated and shown in Fig. A5. It is seen that the sign of ψ shows an oscillation with the particle radius at $\theta_l = 15^\circ$. This indicates that the sign of the dipole moment term and the LOF will reverse as the particle size increases.

The other contribution may originate from the interaction between the higher-order polarons of the particle, since the higher-order multipole is excited as shown by the near field scattering field in Fig. 1 and Fig. 2. Notice that in the dual-dipole model only considers the first-order dipole scattering interaction and ignores the higher-order multipole interaction. The rigorous investigation should be particularly carried out in detail in other work.

Figure A5. The dependence of the ψ on the particle radius R at incident angle $\theta_l = 15^\circ$.

Comment 3: A video showing the successively tuning of the polarization and the particle movement should be given.

Response: Thank you for this comment. In fact, affected by environmental airflow, temperature, vibration, and other factors on the liquid surface water velocity, oil droplets are prone to escape from the capture trap when moving to both ends of the linear optical trap, making it difficult to achieve continuous tuning of all polarization angles in experiments. Here, we present a video showing the lateral motion of an oil droplet with a radius of $10.5\mu\text{m}$ at polarization angles $\alpha = \pm 30^\circ, \pm 45^\circ, \pm 60^\circ$ and $\pm 75^\circ$, see Supplementary Video 2. At the same time, the lateral displacement and lateral movement speed of the oil droplets under different polarization angles in the video are given in Fig. A6. The video and Fig. A6 show that movement speed could achieve maximum at polarization angle $\alpha = \pm 45^\circ$, confirming the tunability of the polarization.

Figure A6. (a-d) The variations of the lateral displacement with the polarization angle switching circularly between $\alpha = \pm 45^\circ, \pm 60^\circ, \pm 75^\circ$. (e-f) The variations of the lateral velocity with the polarization angle.

Comment 4: The authors simulate PS particles, while use the droplet in the experiment, which is not consistent. Why not using PS particles in the experiment or using droplet in the simulation?

Response: Thank you for your constructive comment. Due to the higher degree of circular symmetry for a spherical PS particle, it is more rigorous to confirm the idea of

polarization-induced MSB and the physical origin of LOF. Meanwhile, it is extremely difficult in the experiment to make the spherical PS particle semi-float at the air-water interface, while the lens-like oil droplet is much easier. Even if spherical particles can float on the liquid surface, there is no way to prove that they are exactly semi-floating. However, the shape of oil droplets on the liquid surface can be strictly calculated through the three-phase interfacial tension. Furthermore, lens-like oil droplets are widely used in various applications, such as medical diagnosis, immunoassay, and analytical chemistry. Flexible and precise controllable manipulation of oil droplets is crucial. Therefore, we demonstrated the controllable lateral movement of oil droplets under the action of LOF in our experiments.

In our work, we have used the 3D-Raytracing method in the supplementary material to calculate the LOF of oil droplets of geometric optical size under different polarization angles and incident angles, as shown in Fig. S9 and Fig. S10. The calculated results show that the LOF of the large-sized oil droplets and PS particles semi-floating at the two media exhibits the same polarization dependence, $F_y \propto \sin(2\alpha)$. This is consistent with the result that spherical Rayleigh particles and Mie particles are semi-suspended at the air-water interface. To demonstrate that this polarization-dependent MSB mechanism is applicable to lens-like oil droplets with arbitrary sizes, we further simulated the LOF of a lens-like oil droplet with a radius of 500 nm through the 3D-FDTD method, as shown in Fig. A7. Figure A7a-A7d shows the scattering near-field $|E_z|$ in the yz -plane ($x=0$) at different polarization angle α . The incident angle is 55° . It is seen clearly that the mirror symmetry of the scattering near-field varies with the polarization orientation of the incident light. In Figs. A7a and A7c, when illuminated by the incident plane wave with a polarization angle $\alpha=-45^\circ$ (45°), the scattering near-field $|E_z|$ exhibits an obvious MSB. The scattering near-fields in Fig. A7b and A7d confirm that the light scattering turns back to be mirror-symmetrical and the LOF vanishes when the polarization is tuned to p - ($\alpha=90^\circ$) or s -polarization ($\alpha=0^\circ$). Figure A6e shows the dependence of the normalized LOF of an oil droplet with a radius of 500 nm on polarization angle at different incident angles. The results show that the LOF of oil droplets is proportional to $\sin(2\alpha)$, which is consistent with the 3D-Raytracing

calculation results and experimental results. Figure A7f shows the dependence of the normalized LOF of the oil droplet on the incident angle. The variation of LOF with incident angle is consistent with the 3D-Raytracing calculation results, and the LOF reaches its maximum near $\theta_i=60^\circ$. The above results indicate that lens-like oil droplets, as a special case of the $SO(2)$ rotationally symmetric system, can be used to experimentally verify the proposed novel LOF. The numerical simulation results of Fig. A7 are also added to the supplementary material.

Figure A7. The normalized LOF of an oil droplet with a three-phase contact line radius of 500 nm. (a-d) The scattering near-field $|E_z|$ in the yz -plane ($x=0$) at different polarization angle α . The incident angle is 55° . (e) Dependence of the normalized LOF of the oil droplet on polarization angle at different incident angle. (f) Dependence of the normalized LOF of the oil droplet on incident angle at different polarization angle.

Comment 5: The feasibility of Maxwell stress tensor used in an interface system is not classified.

Response: Thank you for your constructive comment. We verified the effectiveness of MST used in interface systems through the 3D-FDTD method and the full vector finite element (FVFEM) method, respectively, as shown in Fig. A8. Figures A8a and A8b show the normalized LOF calculated using MST in 3D-FDTD and FVFEM respectively.

It can be seen that the LOF obtained by the two simulation methods has almost the same magnitude and trend as the incident angle. It is noted that the numerical results calculated by the FVFEM method are slightly smaller than those calculated by the 3D-FDTD method due to the different grid sizes used in the two calculation methods. In order to reduce the computational memory, the mesh size was set to $\lambda/8/n$ in the FVFEM simulation, λ is the wavelength of the incident light, and n is the refractive index of the material. For a PS particle ($n=1.5983@532\text{nm}$), the mesh size is approximately 40nm. However, the mesh size was set to $R/25$ for particles with a radius R in the 3D-FDTD simulation. For a $1\mu\text{m}$ PS particle, the mesh size is 20nm.

Figure A8. (a) The normalized LOF is calculated by MST in 3D-FDTD method. (b) The normalized LOF is calculated by MST in FVFEM method.

Comment 6: The manuscript needs an extensive revision. It seems that the writing is quite rush. Lots of English writing problems exhibit inside. In light of those issues, I am afraid that I cannot recommend its publication in its current form. It may be considered by a specific optics journal after a thorough revision.

Response: Thank you for your critical and constructive comment. We carefully read the manuscript, corrected incorrect expressions and English writing, and improved the rigor and readability of the manuscript. All of the changes are highlighted with red and underlined marks in the revised manuscript.

Additionally, we believe that the novel MSB mechanism we proposed is innovative and universal. It will provide a novel optical force generation mechanism for many optical manipulation fields and expand new dimensions of optical force regulation.

Therefore, we believe that our work is an important milestone and is very suitable for publication in *Nature Communications*. Regarding the problem of English expression, we will work hard to revise it until it meets the journal standards.

REVIEWER COMMENTS

Reviewer #1 (Remarks to the Author):

I read all the referee reports and the authors' responses. In my opinion, the formal changes in the manuscript did not considerably improve it. I still think that the title is unclear and does not reflect the actual content of this work. I would recommend publication in a more specialized journal, but if other referees support publication in Nature Communication, I do not mind.

Reviewer #2 (Remarks to the Author):

Thanks for the detailed response of authors to the manuscript 'Dynamics of polarization-tuned mirror symmetry breaking in a rotationally symmetric system'. After reviewing the revised version of manuscript, the authors respond clearly to my former concerns: 1. The novelty in contrast to Janus particle, 2. The fundamental physical connections between the mirror symmetric broken and lateral optical force, 3. The applications. We thought the authors replied most of our concerns properly, i.e., 1. The Janus particle is a special case of the SO(2) rotational symmetry system with an absorptive hemisphere, and also the physical mechanism of linear polarization-induced MSB has not been addressed, 2. The authors defined a degree of mirror symmetry breaking DMSB to reveal the difference of the scattering momentum along the $\pm y$ direction to the total scattering momentum along the y direction and the resulted lateral optical force, 3. the proposed MSB mechanism would not only provide an extra tunable dimension for the LOF but also potentially provide an application in size or geometry sorting. Actually, we are a little dissatisfied to the reply to the fundamental physical connections between the mirror symmetric broken and lateral optical force, because the directional scattering will result in the directional optical force. However, as the authors mentioned, understanding the lateral optical force through the MSB should investigate the Lagrange action of the optical field, particle, and their interactions, it would be a complicated and comprehensive work and may not be performed in the current work. Based on the former considerations, we believe the work uncover a novel and unified mechanism of MSB to generate lateral optical force and can recommend the publish of the manuscript in Nature Communications. Some minor revisions:

1. The minus sign - should be replaced by $-$.
2. The font size of equations 1-3 should be adjusted.
3. The equations should be modulated. As the reviewer 3 mentioned, it seems that the writing is quite rush.
4. The format of references should be also modulated.

Reviewer #3 (Remarks to the Author):

The authors have only addressed part of my comments. There remain some important issues to be solved before it can be considered for publication.

1. Previously, I mentioned the inconsistency of particle property used in the simulation and experiment. Though the authors provide more results with oil droplet in the supplementary material, the simulation in the main text still uses the PS bead.
2. Figures 4e and 4f look like the continuous tuning of polarizations, while the authors mention that it is difficult to achieve experimentally in a video. It is misleading.

REVIEWER COMMENTS

Reviewer #1 (Remarks to the Author):

I read all the referee reports and the authors' responses. In my opinion, the formal changes in the manuscript did not considerably improve it. I still think that the title is unclear and does not reflect the actual content of this work. I would recommend publication in a more specialized journal, but if other referees support publication in Nature Communication, I do not mind.

Reviewer #2 (Remarks to the Author):

Thanks for the detailed response of authors to the manuscript 'Dynamics of polarization-tuned mirror symmetry breaking in a rotationally symmetric system'. After reviewing the revised version of manuscript, the authors response clearly to my former concerns: 1. The novelty in contrast to Janus particle, 2. The fundamental physical connections between the mirror symmetric broken and lateral optical force, 3. The applications. We thought the authors replied most of our concerns properly, i.e., 1. The Janus particle is a special case of the $SO(2)$ rotational symmetry system with an absorptive hemisphere, and also the physical mechanism of linear polarization-induced MSB has not been addressed, 2. The authors defined a degree of mirror symmetry breaking DMSB to reveal the difference of the scattering momentum along the $\pm y$ direction to the total scattering momentum along the y direction and the resulted lateral optical force, 3. the proposed MSB mechanism would not only provide an extra tunable dimension for the LOF but also potentially provide an application in size or geometry sorting. Actually, we are a little dissatisfied to the reply to the fundamental physical connections between the mirror symmetric broken and lateral optical force, because the directional scattering will result in the directional optical force. However, as the authors mentioned, understanding the lateral optical force through the MSB should investigate the Lagrange action of the optical field, particle, and their interactions, it would be a

complicated and comprehensive work and may not be performed in the current work. Based on the former considerations, we believe the work uncover a novel and unified mechanism of MSB to generate lateral optical force and can recommend the publish of the manuscript in Nature Communications.

Some minor revisions:

1. The minus sign - should be replaced by ?.
2. The font size of equations 1-3 should be adjusted.
3. The equations should be modulated. As the reviewer 3 mentioned, it seems that the writing is quite rush.
4. The format of references should be also modulated.

Reviewer #3 (Remarks to the Author):

The authors have only addressed part of my comments. There remain some important issues to be solved before it can be considered for publication.

1. Previously, I mentioned the inconsistency of particle property used in the simulation and experiment. Though the authors provide more results with oil droplet in the supplementary material, the simulation in the main text still uses the PS bead.
2. Figures 4e and 4f look like the continuous tuning of polarizations, while the authors mention that it is difficult to achieve experimentally in a video. It is misleading.

-----Comments and Responds-----

Reviewer #1:

Comments to the Author:

I read all the referee reports and the authors' responses. In my opinion, the formal changes in the manuscript did not considerably improve it. I still think that the title is unclear and does not reflect the actual content of this work. I would recommend publication in a more specialized journal, but if other referees support publication in Nature Communication, I do not mind.

Response: Thank you for your peer review and constructive suggestions on this work, which are crucial for improving the quality of the manuscript. Regarding your concern that the title of the manuscript does not reflect the actual content of this work, we think that the main work of this article is not only to propose a new mirror-breaking mechanism to generate lateral optical force (LOF) in a rotational symmetry system, but more importantly, to experimentally verify this novel LOF. The LOF is measured through a dynamic movement of the particle semi-floating at the air-water interface, where the balanced interaction between the LOF and the water viscous friction results in the lateral movement of the particles. Therefore, we believe the "dynamics" in the title could describe this work more comprehensively, because it not only means the mechanism and principle of the LOF, but also indicates the dynamic movement of the particles under the LOF as the result of the balanced interaction.

Reviewer #2:

Comments to the Author:

Thanks for the detailed response of authors to the manuscript 'Dynamics of polarization-tuned mirror symmetry breaking in a rotationally symmetric system'. After reviewing the revised version of manuscript, the authors response clearly to my former concerns: 1. The novelty in contrast to Janus particle, 2. The fundamental physical connections between the mirror symmetric broken and lateral optical force, 3. The applications. We thought the authors replied most of our concerns properly, i.e., 1. The Janus particle is a special case of the $SO(2)$ rotational symmetry system with an

absorptive hemisphere, and also the physical mechanism of linear polarization-induced MSB has not been addressed, 2. The authors defined a degree of mirror symmetry breaking DMSB to reveal the difference of the scattering momentum along the $\pm y$ direction to the total scattering momentum along the y direction and the resulted lateral optical force, 3. the proposed MSB mechanism would not only provide an extra tunable dimension for the LOF but also potentially provide an application in size or geometry sorting. Actually, we are a little dissatisfied to the reply to the fundamental physical connections between the mirror symmetric broken and lateral optical force, because the directional scattering will result in the directional optical force. However, as the authors mentioned, understanding the lateral optical force through the MSB should investigate the Lagrange action of the optical field, particle, and their interactions, it would be a complicated and comprehensive work and may not be performed in the current work. Based on the former considerations, we believe the work uncover a novel and unified mechanism of MSB to generate lateral optical force and can recommend the publish of the manuscript in Nature Communications. Some minor revisions:

Comment 1: The minus sign - should be replaced by ?.

Response: Thank you for your constructive suggestions. We have modified the representation of the minus signs in the manuscript.

Comment 2: The font size of equations 1-3 should be adjusted.

Response: Thank you for your constructive suggestions. We have readjusted the font size of Equations (1-3) to ensure that they can be displayed clearly on the standard page.

Comment 3: The equations should be modulated. As the reviewer 3 mentioned, it seems that the writing is quite rush.

Response: Thank you for your precious suggestions. We have carefully reviewed the format and adjusted the font size of the equations in the manuscript and supplementary material. Besides, we carefully read the manuscript again, corrected incorrect expressions and improved the English writing as much as possible.

Comment 4: The format of references should be also modulated.

Response: Thank you for your constructive suggestions. The format of all references was rechecked and carefully revised to comply with journal requirements. Finally, we sincerely thank you again for your peer review of this work. Your precious suggestions are of great significance to improve the quality of the manuscript.

Reviewer #3:

Comments to the Author:

The authors have only addressed part of my comments. There remain some important issues to be solved before it can be considered for publication.

Comment 1: Previously, I mentioned the inconsistency of particle property used in the simulation and experiment. Though the authors provide more results with oil droplet in the supplementary material, the simulation in the main text still uses the PS bead.

Response: Thank you for your crucial comment. We want to emphasize that the main novelty of this manuscript is the discovery of the LOF induced by a linearly polarized plane wave that commonly exists in an $SO(2)$ rotationally symmetric system. In particular, the strategy of utilizing the buoyancy at the interface is a special case of the $SO(2)$ rotationally symmetric system that is easier to implement experimentally. Therefore, to understand the novel mechanism of mirror symmetry breaking and LOF, it is more convincing to use spherical particles with the highest degree of circular symmetry instead of lens-shaped droplets with additional asymmetry. On the other hand, it is also very difficult to prove in experiments whether spherical particles are semi-floating at the air-water interface. So, we used the oil droplets semi-floating at the liquid surface instead of the PS spheres in the experiment. The shape of oil droplets on the liquid surface can be strictly calculated through the three-phase interfacial tension. After calculations, we found that the oil droplets are lens-shaped and exactly a special case of the $SO(2)$ rotationally symmetric system, which can also be described approximately by the dual-dipole model. The numerical simulation in sections 6 and 7 (the supplementary material) shows that the LOF of the oil droplet semi-floating on the air-water interface is proportional to $\sin(2\alpha)$, which is consistent with the sinusoidal

dependence of the LOF on the polarization angle for the PS particles semi-floating at the air-water interface and for the dual dipole. As can be seen, the spherical PS particle and the lens-shaped oil droplet are respectively different cases of the $SO(2)$ rotational symmetric system, however, exhibiting the same sinusoidal dependence of the LOF on the polarization angle, that is, $F_y \propto \sin(2\alpha)$. Consequently, the numerical simulation of the PS spherical particle and the experiments using lens-shaped oil droplets together support our conclusion that polarization-induced mirror-symmetry breaking and LOF in $SO(2)$ rotationally symmetric system. To make the manuscript more readable, we insert some sentences to explain the logic more clearly. New statements have been added to page 15.

“The sinusoidal dependence indicates that the oil droplet semi-floating at the interface is another special case of the $SO(2)$ rotationally symmetric system, allowing us to verify the MSB and the LOF safely.”

Comment 2: Figures 4e and 4f look like the continuous tuning of polarizations, while the authors mention that it is difficult to achieve experimentally in a video. It is misleading.

Response: Thank you for pointing out the shortcoming in Figs. 4e and 4f. In Figs. 4e and 4f, the switching polarization occurs only at the moment when the direction of motion changes, that is, at the junction of background colors. During the movement of the particle, the polarization always remains at $\alpha = -45^\circ$ or $\alpha = 45^\circ$. The polarization angle is switched circularly with an average period of ~ 63 s. In the experiment, when the oil droplet moves close to the end of the linear optical trap, the polarization angle is immediately switched to make it move in the opposite direction (see Supplementary Video 1). The duration of the polarization angle change is approximately 1s, which is controlled to avoid oil droplets escaping from the linear optical trap. To avoid the “misleading” pointed out by the referee, we further modified Figures 4e and 4f to make the movement of the droplet more explicit. Additionally, considering the environmental airflow, temperature, vibration, and other factors on the liquid surface, the oil droplets are prone to escape from the capture trap when moving to the two ends of the linear

optical trap. Therefore, in a short period, we can implement the periodic back-and-forth motion of oil droplets by continuously switching between the positive and negative polarization angles, such as $\alpha = \pm 45^\circ$, $\pm 60^\circ$ and $\pm 75^\circ$. Experiments show that LOF could indeed be tuned by the polarization angle, which can be verified by the movement speed of oil droplet at the different polarization angles $\alpha = \pm 45^\circ$, $\pm 60^\circ$, $\pm 75^\circ$ (see Supplementary Video 2). As seen in Fig. S13, the LOF at $\alpha = \pm 75^\circ$ and $\pm 60^\circ$ becomes smaller than the LOF at $\alpha = \pm 45^\circ$, and the maximum speed of the oil droplet at $\alpha = \pm 75^\circ$ and $\pm 60^\circ$ is slower than the maximum speed at $\alpha = \pm 45^\circ$. Some new statements have been added to page 16.

“Figure 4b shows the snapshots of the lateral movement of an oil droplet with a three-phase contact line radius $r = 9.7 \mu\text{m}$ when the polarization angle α is periodically switched between -45° and 45° (see Supplementary Movie 1). To avoid oil droplets escaping from the linear optical trap, the polarization angle is switched circularly with an average period of $\sim 63\text{s}$.”

We hope our responses address your concerns. Additionally, we sincerely thank you for your careful review, your constructive comments have indeed helped us improve the quality of the manuscript.

REVIEWERS' COMMENTS

Reviewer #3 (Remarks to the Author):

The authors provide some convincing videos for their demonstrations. I now recommend its publication.